# Quantifying Local Model Validity using Active Learning

**Sven Lämmle**[1, 2]       **Can Bogoclu**[3]       **Robert Voßhall**[4]       **Anselm Haselhoff**[5]       **Dirk Roos**[2]

[1]Center of Expertise, ZF Friedrichshafen AG, Friedrichshafen, Germany
[2]Institute of Modelling and High-Performance Computing, Niederrhein University of Applied Sciences, Krefeld, Germany
[3]Zalando SE, Berlin, Germany
[4]auxmoney GmbH, Düsseldorf, Germany
[5]Ruhr West University of Applied Sciences, Bottrop, Germany

## Abstract

Real-world applications of machine learning models are often subject to legal or policy-based regulations. Some of these regulations require ensuring the validity of the model, i.e., the approximation error being smaller than a threshold. A global metric is generally too insensitive to determine the validity of a specific prediction, whereas evaluating local validity is costly since it requires gathering additional data. We propose learning the model error to acquire a local validity estimate while reducing the amount of required data through active learning. Using model validation benchmarks, we provide empirical evidence that the proposed method can lead to an error model with sufficient discriminative properties using a relatively small amount of data. Furthermore, an increased sensitivity to local changes of the validity bounds compared to alternative approaches is demonstrated.

## 1  INTRODUCTION

Ensuring the validity of deployed machine learning (ML) models is often a core concern in safety-critical domains such as medical, vehicle, and industrial applications, with a high risk of harming humans and the environment. These use cases are often subject to legal or regulatory requirements, such as ISO26262 [2016] and IEC61508 [2010]. Most ML models are built using only past observations or examples and may lack further domain-specific inductive biases, such as underlying physics (although some models are capable of incorporating such, e.g., see [Karniadakis et al., 2021]). Consequently, the behavior of these models in unseen scenarios is difficult to predict without further analysis.

A strict assessment of a model's capabilities is needed to determine its validity across the input space. Specifically, we want to identify valid subdomains of the input space where the absolute model error is smaller than some predefined tolerance level. In this context, evaluating global accuracy metrics such as mean squared error is not a useful approach. Even if a model achieves a small average error globally, it can exhibit high inaccuracy in specific input domains. Similarly, even models with high average error may be useful in certain subdomains of the input space.

The validity of a prediction can be assessed by comparing it to real-world observations or, in some cases, to simulations with very high accuracy. The obtained results can be used to approximate the error level [Oberkampf and Roy, 2010], sometimes referred to as *error learning* [Riedmaier et al., 2020]. In contrast to boosting, we are not interested in improving the model predictions by the addition of the learned error. Instead, we want to have an estimate of the error level to decide the validity of a prediction. A popular model for this task is Gaussian process regression (GP) [Rasmussen and Williams, 2006, Kennedy and O'Hagan, 2001]. Besides being a powerful model, its capability to represent the epistemic uncertainty is useful to derive confidence bounds for the estimated local error.

If a large dataset is available for training, an additional split can be afforded to build an error model to be used for model validation (MV). However, gathering additional data is costly but necessary in many cases to build a sufficiently accurate validation model. Therefore, the *design of experiments* (DoE), i.e., the planning of tests or data queries, is often crucial as it determines the overall validation cost and the quality of the validation statement. A good strategy should achieve a highly accurate validity estimate while being data-efficient, using as little additional data as possible.

Active learning [Settles, 2010] has shown to be an efficient strategy for reducing the number of samples, i.e., the choice of samples used for training a model. Therefore, samples or batches of samples are selected sequentially, leveraging knowledge from previous iterations to guide the sampling

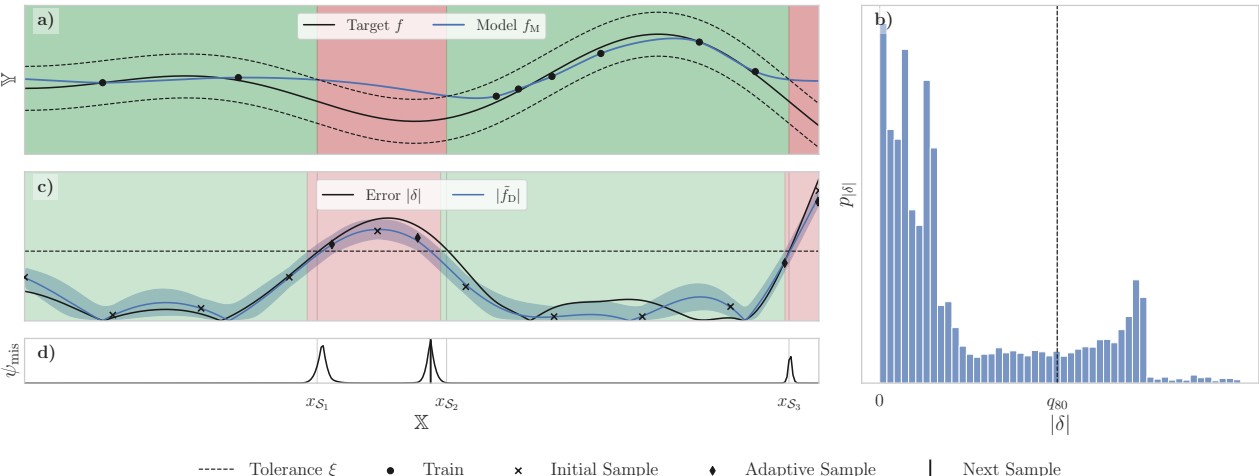

Figure 1: Illustration of a locally valid model: the trained model $f_M$ is marginally valid for tolerance level $\xi$ with $80\%$ probability (**b**)), but only locally valid $\mathcal{V}$ (■) in some regions of the input space $\mathbb{X}$ (**a**)). **b)** Marginal distribution of the true absolute error $|\delta|$, where the $80\%$ quantile corresponds to the tolerance level, i.e., $\xi = q_{80}$. **c)** Our learned error model $|\tilde{f}_D|$ and $90\%$ confidence interval (■) from the folded Gaussian (Section 4.3), together with the predicted local valid set $\tilde{\mathcal{V}}_{0.1}$ (●) (Section 4.6). Samples (♦) are sequentially placed based on $\psi_{\text{mis}}$ (**d**)) to reduce the misclassification probability (Section 4.4), i.e., most samples are close to the limit state $x_{\mathcal{S}_i} \in \mathcal{S}$.

process. This approach has been used across various tasks, from optimization to querying new samples [Kumar and Gupta, 2020] for improving model quality.

As an active learning method, Bayesian optimization (BO) [Snoek et al., 2012] is known to exploit the probabilistic estimates to find the optimum of a black-box objective function with high data efficiency. However, our goal is to find valid domains, i.e., the set $\mathcal{V}$ of points $\mathbf{x}$ where the absolute model error $|\delta(\mathbf{x})| \in \mathbb{R}_+$ is smaller than some tolerance $\xi \in \mathbb{R}_{>0}$. In contrast, global adaptive sampling strategies [Lämmle et al., 2023] aim to improve the model quality over the entire input space. We are not interested in having accurate predictions where the error $|\delta(\mathbf{x})|$ is much larger or smaller than $\xi$. Instead, we are primarily interested in learning the neighborhood where $|\delta(\mathbf{x})| \approx \xi$. In classical engineering, reliability analysis (RA) requires solving a similar problem [Rebba and Mahadevan, 2008].

Given a function $g(\cdot)$, RA defines failure domains using the condition $g(\mathbf{x}) \leq 0$. The aim of RA is to compute the failure probability $P_{\mathcal{F}} = \mathbb{E}_{\mathbf{X} \sim p(\mathbf{x})} \left[ \mathbb{1}_{g(\mathbf{X}) \leq 0}(\mathbf{X}) \right]$, where $\mathbf{X}$ are the input variables. Therefore, an accurate representation of the *limit state* condition $g(\mathbf{x}) = 0$ is necessary. Equivalently, the problem of MV can be framed as learning two *level sets* [Gotovos et al., 2013] at once: $\mathcal{V} = \{\mathbf{x} \colon \delta(\mathbf{x}) \leq \xi\} \cap \{\mathbf{x} \colon \delta(\mathbf{x}) \geq -\xi\}$. Despite the similarity of the problems and the popularity of active learning in RA and level set estimation, to the best of our knowledge, a similar approach has not been proposed for MV.

In this paper, we first formulate the problem of MV as

learning two limit state conditions ($\delta(\mathbf{x}) = \xi$ and $\delta(\mathbf{x}) = -\xi$), thus showing the connection to RA problems. Based on this formulation, we propose an active learning method for MV inspired by its RA counterpart Bichon et al. [2008]. We test our method on a variety of benchmark problems within a small sample setting and show that it can be reliably used for the validation of a multitude of ML models.

**Contributions.** Our main contributions are summarized as follows: **1)** We introduce a new formulation of model validation (MV) inspired by reliability analysis (RA), extending the setting to two symmetrical limit state conditions and noisy observations. **2)** We propose a novel acquisition function based on the misclassification probability of the limit state (Section 4.4). **3)** We derive frequentist error bounds for the proposed methodology (Section 4.7). **4)** We evaluate the proposed method on a variety of different benchmarks (Section 5, Appendix E), and provide a comparison with conformal prediction (Appendix D).

## 2 RELATED WORK

**Reliability Analysis.** The goal of RA is to calculate the failure probability $P_{\mathcal{F}}$. Since the distribution of $\mathbf{X}$ can be arbitrary, RA often requires a large number of samples. To increase sample efficiency, previous research has utilized surrogate-aided methods based on ML models such as neural networks [Papadrakakis et al., 1996], ordinary [Bucher and Bourgund, 1990] and moving least squares models [Most and Bucher, 2006], GPs [Kaymaz, 2005], and

support vector machines [Rocco and Moreno, 2002]. Although some active learning approaches have been already proposed [Macke et al., 2000, Most and Bucher, 2006], Bichon et al. [2008] was the first to introduce active learning based on GP models to the field of RA.

Following, several acquisition functions (AFs) similar to Bayesian optimization methods have been proposed to learn the limit state. The most popular ones being the *expected feasibility function* [Bichon et al., 2008] and the *U-function* [Echard et al., 2011, 2013]. For an extensive review, see [Teixeira et al., 2021]. In contrast, active learning in the ML context aims to achieve high accuracy *globally* by using as few samples (labels) as possible. The approach is more frequently used in the context of classification tasks compared to regression. See [Settles, 2010] for an overview.

**Level Sets.** Similar to RA, the task of level set estimation [Bryan et al., 2005, Gotovos et al., 2013] aims to identify regions where the value of some target function is below or above a given threshold. Therefore, samples are placed according to an AF based on a GP model to reduce the uncertainty about the level set. In this context, MV can be seen as learning two joint level sets simultaneously. A more general approach, not restricted to level set estimation, was proposed by Neiswanger et al. [2021]. The entropy-based method can be applied to arbitrary algorithmic outputs. However, the computational cost for evaluating the AF may be high, as a closed-form is unavailable.

**Bayesian Calibration.** Another line of work initiated by Kennedy and O'Hagan [2001] considers learning the model error to correct the output of a computational model similar to boosting techniques in ML, while inferring the free parameters of the computational model. Therefore, the posterior distribution of the parameters and the model error are inferred jointly. Subsequent research incorporates physical knowledge by using a physics-informed prior [Spitieris and Steinsland, 2023] and extends the work to multiple outputs [Higdon et al., 2008, White and Mahadevan, 2023]. Nevertheless, one mayor criticism of this work is the identifiability issue, i.e., the effects of calibration parameters and model discrepancy can be confounded due to the over-parameterization of the model [Arendt et al., 2012, Marmin and Filippone, 2022].

**Conformal Prediction.** For distribution-free predictive inference, conformal prediction based on the work of Vovk et al. [2005] is used to derive prediction intervals with frequentist coverage guarantees. However, most work focuses on marginal coverage [Vovk et al., 2005, Papadopoulos et al., 2002, Lei et al., 2018], i.e., the model prediction is only marginally valid over the training data and the test points with a specified probability. In this context, our work can be seen as related to the setting of split-conformal prediction, where the data is partitioned into training and calibration sets. The latter is used to derive the prediction intervals for the trained model. Here, we aim to learn the model error, especially around the limit state, to assess the local validity of the model.

A more rigorous conditional coverage is known to be impossible to achieve without further assumptions about the underlying distribution [Barber et al., 2021]. Approximate conditional coverage has been considered in the work of Lin et al. [2021]. Although they provide a method for approximate conditional coverage, it often requires more calibration samples to form meaningful prediction intervals (see Appendix D).

# 3 BACKGROUND

Let $\mathbf{x} \in \mathbb{X} \subset \mathbb{R}^d$ be a vector of observable and controllable inputs, with validation domain $\mathbb{X}$ of dimension $d$. Furthermore, for a model represented by a map $f_{\mathrm{M}} \colon \mathbb{X} \to \mathbb{Y}$ between the inputs $\mathbb{X}$ and the quantity of interest (QOI) $\mathbb{Y} \subset \mathbb{R}$, we want to assess the validity of $f_{\mathrm{M}}$, i.e., estimate model misfit.

**Observations.** An observation consists of the tuple $(\mathbf{x}, y_x)$ where $y_x$ is assumed to be a sample from the random variable $Y_{\mathbf{x}} = f_{\mathrm{E}}(\mathbf{x}) + \epsilon$. In this context, $f_{\mathrm{E}} \colon \mathbb{X} \to \mathbb{Y}$ is the true data generating function and $\epsilon$ is the additive label noise. Here, we do not assume to have any knowledge about $f_{\mathrm{E}}$, therefore we treat it as a black-box function. Moreover, $\epsilon$ is assumed to be homoscedastic Gaussian white noise $\epsilon \sim \mathcal{N}(0, \sigma_{\mathrm{e}}^2)$, where we use $\sim$ to denote "distributed as" in this work. We assume that the input can be measured precisely, such that any uncertainty about the true value of $\mathbf{x}$ can be neglected.

In general, a set of observations is used to train and possibly calibrate the model $f_{\mathrm{M}}$. Another set of observations is used to estimate the generalization error and to validate $f_{\mathrm{M}}$. Generating new observations is often costly, and only a finite number of observations are available in real-world applications. Therefore, data efficiency of both training and validation is crucial.

**Validation Metric.** MV can be performed by evaluating the model error within $\mathbb{X}$, given as $f_{\mathrm{D}}(\mathbf{x}) := f_{\mathrm{M}}(\mathbf{x}) - Y_{\mathbf{x}}$, with $f_{\mathrm{D}}(\mathbf{x}) \sim \mathcal{N}(f_{\mathrm{M}}(\mathbf{x}) - f_{\mathrm{E}}(\mathbf{x}), \sigma_{\mathrm{e}}^2)$. For a given tolerance level $\xi \in \mathbb{R}_{>0}$, we quantify the probability of the model error being within the desired tolerance as

$$P\left(-\xi < f_{\mathrm{D}}(\mathbf{x}) < \xi\right). \tag{1}$$

Equation (1) is termed the *reliability validation metric* and was proposed for MV by Rebba and Mahadevan [2008], Sankararaman and Mahadevan [2013]. Equivalently, we will represent Equation (1) as $P\left(g(\mathbf{x}) > 0\right) = 1 - P\left(g(\mathbf{x}) \leq 0\right)$, where we have $g(\mathbf{x}) := \xi - |f_{\mathrm{D}}(\mathbf{x})|$, which

is in reliability theory often referred to as *limit state function*. In the following, we will define the noiseless residual as $\delta(\mathbf{x}) = \mathbb{E}\left[f_\mathrm{D}(\mathbf{x})\right] = f_\mathrm{M}(\mathbf{x}) - f_\mathrm{E}(\mathbf{x})$, which is unknown in practice, i.e., only $\xi$ and $f_\mathrm{D}(\mathbf{x})$ are available for validation iff. $\mathbf{x}$ is already observed input.

**Differences to RA.** The formulation of MV leads to key differences to RA that we have to consider: **1)** The formulation of Equation 1 is analogous to having two limit state conditions in the setting of RA. **2)** The distribution of the samples in the limit state is a folded Gaussian [Leone et al., 1961], since $g(\cdot)$ is formulated in terms of the absolute value of $f_\mathrm{D}(\cdot)$, which is Gaussian by assumption. **3)** The MV limit state is corrupted by noise, while it is commonly considered noise-free in RA[1], since $g(\cdot)$ is generally a simulation model with negligibly small numerical errors [Bucher and Bourgund, 1990]. **4)** $\mathbf{x}$ is subject to uncertainty in RA problems, whereas we assumed no uncertainty in the context of MV. Nevertheless, the MV setting could be interpreted as a RA problem in this respect, where $p(\mathbf{x})$ is a uniform distribution defined over the entire $\mathbb{X}$. This represents our interest in assessing the validity of the model everywhere in $\mathbb{X}$ with equal importance.

# 4 METHOD

In this section, we derive our formulation for learning and representing the limit state $\mathcal{S}$. We start by defining local validity and the limit state for MV. Next, we introduce the notation and active learning approach. In Section 4.3, we show how to represent the limit state function $g(\cdot)$ as a transformed GP model, which is then used in Section 4.4 to derive the AF. A stopping criterion is proposed in Section 4.5 based on the probability of misclassifying the validity of a set of points. Additionally, in Section 4.6 it is shown how to obtain the prediction of the local valid set. Finally, theoretical considerations are presented in Section 4.7.

## 4.1 DEFINITION OF LOCAL VALIDITY AND LIMIT STATE

Before describing the proposed method, proper definitions of local validity and the limit state for the *noiseless* case (i.e., representing the ground truth) are provided.

**Definition 1** (Local Validity). *A model $f_\mathrm{M}$ is locally valid at $\mathbf{x} \in \mathbb{X}$, given a tolerance level $\xi$, if $\xi - |\delta(\mathbf{x})| \geq 0$. Then, the valid region of $f_\mathrm{M}$ is*

$$\mathcal{V} = \{\mathbf{x} \in \mathbb{X}\colon \xi - |\delta(\mathbf{x})| \geq 0\}.$$

Based on the definition of local validity, *global validity* can be asserted if Definition 1 holds for all $\mathbf{x} \in \mathbb{X}$. Note the

difference between global validity and valid on average, e.g., according to a prediction accuracy metric.

**Definition 2** (Limit State). *Given a model $f_\mathrm{M}$ and a tolerance level $\xi$, the limit state of $f_\mathrm{M}$ is*

$$\mathcal{S} = \{\mathbf{x} \in \mathbb{X}\colon \xi - |\delta(\mathbf{x})| = 0\}.$$

We can interpret Definition 2 as the set of points on the boundary between the valid and invalid domains (see Figure 1). Moreover, $\mathcal{S}$ is unavailable in practice, and our objective is to construct a strategy aimed at placing samples in the vicinity of $\mathcal{S}$, thereby efficiently learning to differentiate between valid and invalid domains for $f_\mathrm{M}$.

## 4.2 OVERVIEW

**Notation.** We consider learning the limit state $\mathcal{S}$ over the *normalized* input space $\tilde{\mathbb{X}} \subseteq [0,1]^d$ from validation data $\mathcal{D} = \{(\mathbf{x}_i, y_i)\}_{i=1}^n$, with input $\mathbf{x}_i \in \tilde{\mathbb{X}}$ and label $y_i = f_\mathrm{D}(\mathbf{x}_i)$. Equivalently, we represent training examples as $n \times d$ matrix $\mathbf{X}$, where the $i$-th row is the $i$-th training example $\mathbf{x}_i$, with corresponding labels $\mathbf{y}$. Furthermore, we employ a surrogate model $\hat{g}(\cdot)$ that provides, for some input $\mathbf{x}$, a (conditional) probability distribution over the output $\hat{g}(\mathbf{x}) = \hat{G}_\mathbf{x} \sim p(g|\mathbf{x}, \mathcal{D})$. We can further use, e.g., the predictive mean $\mu_{g|\mathcal{D}}(\mathbf{x}) = \mathbb{E}[\hat{G}_\mathbf{x}]$ or variance $\sigma^2_{g|\mathcal{D}}(\mathbf{x}) = \mathbb{V}[\hat{G}_\mathbf{x}]$ of the model.

**Active Learning.** Active learning strategies aim to reduce the evaluations of an expensive black-box function $f_\mathrm{E}$, while still achieving a satisfactory result; in this context, learning the limit state with high accuracy for MV. Therefore, an *acquisition function* (AF) $\psi\colon \tilde{\mathbb{X}} \to \mathbb{R}$ is used to rate promising new sample points, often referred to as candidates. Learning is encouraged by maximizing $\psi$ over the candidate set $\mathcal{C} = \{\mathbf{c}_i\}_{i=1}^{n_c}$, where $\mathbf{c}_i$ is drawn uniformly from $[0,1]^d$ in our case. A new query $\mathbf{x}^*$ is obtained as

$$\mathbf{x}^* = \underset{\mathbf{x} \in \mathcal{C}}{\arg\max}\, \psi(\mathbf{x}; \hat{g}, \mathcal{D}),$$

where $\hat{g}$ is the learned surrogate based on $\mathcal{D}$. The initial dataset can be generated from a space-filling design (e.g., Sobol [Joe and Kuo, 2008] or Latin hypercube sampling (LHS) [McKay et al., 1979]). Algorithm 1 shows the active learning procedure. From here on, we drop writing the explicit dependence of $\psi$ on $\mathcal{D}$ and $\hat{g}$.

## 4.3 GAUSSIAN PROCESS

Gaussian process regression (GP) is a popular probabilistic ML method, which can be used to represent the belief over the objective function. Therefore, it is a central component to different learning schemes, e.g., in BO [Snoek et al.,

---

[1]With a suitable GP model, we could extend RA to the noisy setting if required (e.g., [Chun, 2024]).

**Algorithm 1** Active Learning Model Validation

---

**Require:** Initial data $\mathcal{D}$, candidate set $\mathcal{C}$, tolerance $\xi$, and acquisition function $\psi$

  **repeat**

      Train surrogate $\hat{g}$ with $\mathcal{D}$

      $\mathbf{x}^* \leftarrow \arg\max_{\mathbf{x}\in\mathcal{C}} \psi(\mathbf{x}; \hat{g}, \mathcal{D})$

      $y^* \leftarrow f_{\mathrm{D}}(\mathbf{x}^*)$

      $\mathcal{D} \leftarrow \mathcal{D} \cup \{(\mathbf{x}^*, y^*)\}$

      Generate new $\mathcal{C}$             ▷ optional

  **until** Termination condition  ▷ e.g., maximum iterations

  Train surrogate $\hat{g}$ with $\mathcal{D}$

  **return** $\mathcal{D}, \hat{g}$

---

2012] or global improvement of surrogate models [Lämmle et al., 2023].

In the following, we use a transformed GP model to represent our belief of the limit state

$$\hat{g} = \lambda \circ \tilde{f}_{\mathrm{D}}$$
$$\tilde{f}_{\mathrm{D}} \sim \mathcal{GP}(\mu, k),$$

where $\circ$ is the function composition, $\mu \colon \tilde{\mathbb{X}} \to \mathbb{R}$ and $k \colon \tilde{\mathbb{X}} \times \tilde{\mathbb{X}} \to \mathbb{R}_+$ denote the *mean* and the *covariance* (kernel) functions, respectively. $\lambda(\cdot)$ represents the non-invertible mapping $\lambda(y) := \xi - |y|$.

**Exact Prediction.** Since the GP is defined as a joint Gaussian distribution, the prediction at a point $\mathbf{x}^\star$ can be analytically obtained as conditional Gaussian distribution $p(y|\mathbf{x}^\star, \mathcal{D}) \sim \mathcal{N}(\mu_{y|\mathcal{D}}(\mathbf{x}^\star), \sigma^2_{y|\mathcal{D}}(\mathbf{x}^\star))$, with mean and variance as

$$\mu_{y|\mathcal{D}}(\mathbf{x}^\star) = \mu(\mathbf{x}^\star) + \mathbf{k}^T(\mathbf{K}^{-1} + \hat{\sigma}^2_{\mathrm{e}}\mathbf{I})\mathbf{y}, \qquad (2)$$

$$\sigma^2_{y|\mathcal{D}}(\mathbf{x}^\star) = k(\mathbf{x}^\star, \mathbf{x}^\star) - \mathbf{k}^T(\mathbf{K}^{-1} + \hat{\sigma}^2_{\mathrm{e}}\mathbf{I})\mathbf{k}, \qquad (3)$$

where $\mathbf{K}$ contains all pairs of kernel entries (i.e., $\mathbf{K}_{ij} = k(\mathbf{x}_i, \mathbf{x}_j)$), and $\mathbf{k}$ denotes the vector of correlations between $\mathbf{x}^\star$ and training points, $\mathbf{k}_i = k(\mathbf{x}_i, \mathbf{x}^\star)$. $\hat{\sigma}^2_{\mathrm{e}}$ is the estimated noise variance.

**Learning Hyperparameters.** Predictions made with a GP depend on the *hyperparameters* $\hat{\sigma}^2_{\mathrm{e}}$ and $\boldsymbol{\theta}$, e.g., noise variance, kernel lengthscale, or possibly the parameters of the mean function $\mu(\mathbf{x}^\star)$. These hyperparameters could be obtained by maximizing the log *marginal likelihood*

$$\log p(\mathbf{y}|\mathbf{X}, \boldsymbol{\theta}, \hat{\sigma}^2_{\mathrm{e}}) \propto -\frac{1}{2}\log|\mathbf{K}| - \frac{1}{2}\mathbf{y}^T(\mathbf{K}^{-1} + \hat{\sigma}^2_{\mathrm{e}}\mathbf{I})\mathbf{y}.$$

If prior knowledge is available, it can be beneficial to use instead the *maximum a posteriori* (MAP) estimate as

$$\hat{\boldsymbol{\theta}}_{\mathrm{MAP}} = \arg\max_{\boldsymbol{\theta}, \hat{\sigma}^2_{\mathrm{e}}} \log p(\mathbf{y}|\mathbf{X}, \boldsymbol{\theta}, \hat{\sigma}^2_{\mathrm{e}}) + \log p(\boldsymbol{\theta}, \hat{\sigma}^2_{\mathrm{e}}),$$

where $p(\boldsymbol{\theta}, \hat{\sigma}^2_{\mathrm{e}})$ are the specified priors over the hyperparameters.

**Limit State Prediction.** The prediction of the limit state is obtained by the mapping $\lambda(\cdot)$. We can give closed form solutions for $\mu_{g|\mathcal{D}}(\cdot)$ and $\sigma^2_{g|\mathcal{D}}(\cdot)$, since $\hat{G}_{\mathbf{x}}$ is a folded Gaussian distribution flipped and shifted by $\xi$ (see Figure 4). Therefore, we have

$$\mu_{g|\mathcal{D}}(\mathbf{x}^\star) = \xi - \left(\sigma_\star\sqrt{\frac{2}{\pi}}\zeta + \mathrm{erf}\left(\frac{\mu_\star}{\sqrt{2\sigma^2_\star}}\right)\mu_\star\right),$$

$$\sigma^2_{g|\mathcal{D}}(\mathbf{x}^\star) = \mu^2_\star + \sigma^2_\star - \mu^2_{g|\mathcal{D}}(\mathbf{x}^\star),$$

where we denoted $\mu_\star = \mu_{y|\mathcal{D}}(\mathbf{x}^\star)$, $\sigma^2_\star = \sigma^2_{y|\mathcal{D}}(\mathbf{x}^\star)$, and $\zeta = \exp\left(\frac{-\mu^2_\star}{2\sigma^2_\star}\right)$. $\mathrm{erf}(\cdot)$ is the Gaussian error function.

### 4.4 ACQUISITION FUNCTION

The AF is used to guide the sampling strategy in regions of interest. Especially for RA and validation, we are interested to sample in the vicinity of the limit state $\mathcal{S}$. For this purpose, several AFs were proposed in RA, e.g., see Bichon et al. [2008], Echard et al. [2011]. Among them, the so called "U-function" [Echard et al., 2011] is widely used in RA [Dubourg et al., 2013, Teixeira et al., 2021].

**U-Function.** The AF focuses on the design subspace near the limit state boundary considering an exploration-exploitation trade-off. It is given by

$$\psi_{\mathrm{U}}(\mathbf{x}) = -\frac{|\mu_{g|\mathcal{D}}(\mathbf{x})|}{\sigma_{g|\mathcal{D}}(\mathbf{x})}, \qquad (4)$$

where $\mu_{g|\mathcal{D}}(\mathbf{x})$ and $\sigma_{g|\mathcal{D}}(\mathbf{x})$ are predictive mean and standard deviation of the probabilistic model, respectively. $\psi_{\mathrm{U}}$ selects samples which have large variance and are close to the limit state according to the GP. $\mu_{g|\mathcal{D}}(\mathbf{x})$ is small only if $\mathbf{x}$ is close to the limit state. Note that Equation 4 is different from the original formulation [Echard et al., 2011] in our setting, since the GP predictive mean and standard deviation are transformed by $\lambda(\cdot)$.

**MC-Prob.** Maximizing $\psi_{\mathrm{U}}$ was originally derived to be equivalent to maximizing the probability of misclassifying the limit state condition under the Gaussian assumption [Echard et al., 2013]. However, the equivalence does not hold for a folded Gaussian posterior. Instead, we propose using the misclassification probability directly as

$$\psi_{\mathrm{mis}}(\mathbf{x}; \omega) = \begin{cases} P\left(\hat{G}_{\mathbf{x}} \leq -\omega\right), & \text{for } |\mu_{y|\mathcal{D}}(\mathbf{x})| \leq \xi \\ 1 - P\left(\hat{G}_{\mathbf{x}} \leq \omega\right), & \text{for } |\mu_{y|\mathcal{D}}(\mathbf{x})| > \xi, \end{cases}$$

where $\omega \in \mathbb{R}_+$ determines the exploration-exploitation trade-off, i.e., it gives the misclassification probability around the limit state with a small slack variable $\omega < \xi$. Larger values of $\omega$ encourage more exploration. The closed-form expression for the cumulative distribution function

(cdf) of $\hat{G}_{\mathbf{x}}$ is given by

$$
\begin{aligned}
P\left(\hat{G}_{\mathbf{x}} \leq \omega\right) = 2 - \Phi\left(\frac{\xi - \omega + \mu_{y|\mathcal{D}}(\mathbf{x})}{\sigma_{y|\mathcal{D}}(\mathbf{x})}\right) \\
- \Phi\left(\frac{\xi - \omega - \mu_{y|\mathcal{D}}(\mathbf{x})}{\sigma_{y|\mathcal{D}}(\mathbf{x})}\right),
\end{aligned}
\tag{5}
$$

where $\Phi(\cdot)$ is the standard normal distribution. See Appendix A for a derivation of Equation 5 as well as Appendix C.2 for some additional insights on the misclassification probability.

## 4.5 STOPPING CRITERION

The criterion for stopping the sequential strategy is a crucial part of the method. One straightforward approach is to use the probability of misclassifying a candidate point $\mathbf{x}$ under the transformed GP posterior, as given by $\psi_{\mathrm{mis}}$. Then, we stop by reaching a predefined tolerance $\alpha \geq \tilde{P}_{\mathrm{mis}} = \mathbb{E}_{\boldsymbol{X} \sim p(\mathbf{x})}\left[\psi_{\mathrm{mis}}(\boldsymbol{X}; \omega = 0)\right]$. In practice, we estimate this expectation based on the candidate set $\mathcal{C}$. The criterion can be made more robust by ensuring the above condition in $k$ consecutive iterations. Additionally, we introduce a sampling budget, i.e., we limit the maximum number of samples obtained. Therefore, we stop if the budget is exhausted or a suitable $\tilde{P}_{\mathrm{mis}}$ is achieved in time.

## 4.6 PREDICTION

The learned GP model represents our final belief of the limit state, which we can use to decide if $f_{\mathrm{M}}$ is locally valid at an arbitrary $\mathbf{x} \in \tilde{\mathbb{X}}$. Hence, we predict the local valid set, analogous to Definition 1, as

$$
\tilde{\mathcal{V}} = \left\{\mathbf{x} \in \tilde{\mathbb{X}}: \xi - |\mu_{y|\mathcal{D}}(\mathbf{x})| \geq 0\right\}.
\tag{6}
$$

In safety-critical applications, preventing false positives can become more important than incorrectly classifying a sample to be invalid, as also noted by Reeb et al. [2023]. Therefore, depending on the application, we may be more risk averse than using $\tilde{\mathcal{V}}$ directly. Instead, we can derive the predicted local valid set with $1 - \alpha$ confidence, for $\alpha \in (0, 1)$, as

$$
\tilde{\mathcal{V}}_{\alpha} = \left\{\mathbf{x} \in \tilde{\mathbb{X}}: q_{\alpha}(\mathbf{x}) \geq 0\right\},
\tag{7}
$$

with quantile $q_{\alpha}(\mathbf{x}) := \inf\{\tilde{g} \in \mathbb{R} : P(\hat{G}_{\mathbf{x}} \leq \tilde{g}) \geq \alpha\}$.

It can be seen from Equation (6) that $\xi$ can be changed post-hoc. However, if $|\xi_{\mathrm{old}} - \xi_{\mathrm{new}}|$ is large, the model may not be sufficiently accurate, as the samples are usually placed near $\xi_{\mathrm{old}}$, and the new limit state resulting from $\xi_{\mathrm{new}}$ could be far away from the old one.

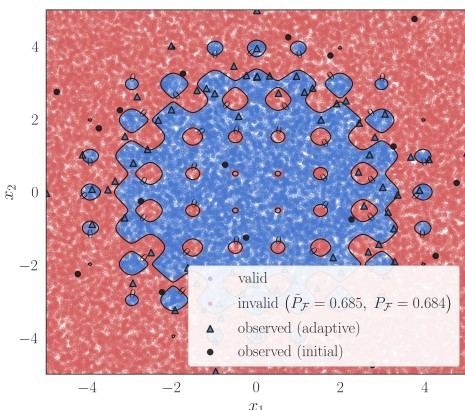

Figure 2: Prediction $\tilde{\mathcal{V}}$ (Equation 6) for the modified Rastrigin function after 20 initial and 70 adaptive observations, with $\psi_{\mathrm{mis}}$ and $\omega = 0.2\xi$. The true limit state is represented by the black line.

## 4.7 THEORETICAL CONSIDERATIONS

By [Lederer et al., 2021, Thm 9] ([Lederer et al., 2019, Thm 3.1], resp.), the regression error of the GP model is under certain conditions bounded in terms of the posterior variance:

**Theorem 1.** *Assume that $\delta$ is a Lipschitz continuous sample from the zero mean Gaussian process with covariance kernel $k$ with Lipschitz constant $L_k$ on the compact set $\tilde{\mathbb{X}}$. Denote the Lipschitz constant of $\delta$ by $L_{\delta}$. Then, $\mu_{y|\mathcal{D}}(\cdot)$ and $\sigma^2_{y|\mathcal{D}}(\cdot)$ are continuous with Lipschitz constants $L_{\mu}$ and $L_{\sigma^2}$ on $\tilde{\mathbb{X}}$ such that*

$$
L_{\mu} \leq L_k \sqrt{n}\|\left(\mathbf{K} + \sigma_{\mathrm{e}}^2\mathbf{I}\right)^{-1}\mathbf{y}\|
\tag{8}
$$

$$
L_{\sigma^2} \leq 2L_k\left(1 + n\|\left(\mathbf{K} + \sigma_{\mathrm{e}}^2\mathbf{I}\right)^{-1}\| k^*\right),
\tag{9}
$$

*where $k^* := \max_{\mathbf{x}, \mathbf{x}' \in \mathbb{X}} k(\mathbf{x}, \mathbf{x}')$. Moreover, pick $\alpha \in (0, 1)$, $\tau \in \mathbb{R}_+$ and set*

$$
\beta(\tau) = 2\log\left(\frac{M(\tau, \tilde{\mathbb{X}})}{\alpha}\right)
$$

$$
\gamma(\tau) = (L_{\mu} + L_{\delta})\tau + \sqrt{\beta(\tau)L_{\sigma^2}\tau},
$$

*where $M(\tau, \tilde{\mathbb{X}})$ is the $\tau$-covering number of $\tilde{\mathbb{X}}$. Then, it holds that*

$$
P\left(|\delta(\mathbf{x}) - \mu_{y|\mathcal{D}}(\mathbf{x})| \leq \eta(\mathbf{x}), \forall \mathbf{x} \in \tilde{\mathbb{X}}\right) \geq 1 - \alpha
\tag{10}
$$

*where $\eta(\mathbf{x}) = \sqrt{\beta(\tau)}\sigma_{y|\mathcal{D}}(\mathbf{x}) + \gamma(\tau)$.*

For $\tilde{\mathbb{X}} \subseteq [0, 1]^d$, it holds

$$
M(\tau, \tilde{\mathbb{X}}) \leq M(\tau, [0, 1]^d) = \left(\frac{\sqrt{d}}{2\tau}\right)^d.
$$

In contrast to the mere consideration of $\sigma_{y|\mathcal{D}}$, Theorem 1 implies convergence of $\mu_{y|\mathcal{D}}$ to $\delta$ if $\eta$ converges to zero sufficiently fast as $n \to \infty$. According to the posterior variance bounds in [Lederer et al., 2021, Section 3], this is particularly the case if a sufficient number of samples are close to $\mathbf{x}$. Since we are mainly interested in a small error near the limit state, it is important to choose the adaptive sampling method accordingly. The proposed AF is designed exactly for this purpose; it intuitively prefers samples which are presumably close to the limit state and have high posterior variance. $\eta(\mathbf{x})$ can be computed explicitly and yields an uniform error bound on $\tilde{\mathbb{X}}$. Since

$$|\delta(\mathbf{x})| \leq |\delta(\mathbf{x}) - \mu_{y|\mathcal{D}}(\mathbf{x})| + |\mu_{y|\mathcal{D}}(\mathbf{x})|,$$

these bounds can be considered as an alternative to the use of confidence intervals of the GP. However, Theorem 1 requires additional knowledge on the Lipschitz continuity of the covariance kernel as well as $\delta$ and is therefore not generally applicable. Furthermore, an exemplary computation shows that the obtained results are very conservative, in particular, if the bounds in (8) and (9) are used (see Appendix C.1). Moreover, we found that the GP confidence intervals provide reliable bounds in most cases.

# 5 EXPERIMENTS

Experiments are conducted to demonstrate the sample efficiency of the presented method on a variety of problems, ranging from analytical benchmark functions to trained ML models on analytical problems, and some real-world tabular datasets from `OpenML` [Vanschoren et al., 2013].[2] Results for the latter are shown in Appendix E.1, while the influence of label noise is studied in Appendix E.2, and results for the stopping criterion are presented in Appendix E.4.

**Target.** The learning target for our benchmarks is to correctly predict valid regions $\mathcal{V}$ of the model $f_M$ as shown in Figure 2. Thus, we can frame the task as a binary classification problem (positive class for valid, and negative for invalid) and report precision and recall, where the $F_1$-score is used as a summary within plots. For the ground truth (GT) labels, we evaluate $\xi - |\delta(\mathbf{x})|$, where $\delta(\mathbf{x}) = f_M(\mathbf{x}) - f_E(\mathbf{x})$ is the noiseless residual. Further, we split into valid (positive) GT label if the value is not negative and invalid label otherwise (Definition 1).

**Setup.** The initial, candidate, and test datasets are drawn quasi-uniform via LHS, with $10d$ initial observations as proposed by Loeppky et al. [2009], where $d$ is the dimension of the validation domain. We restrict the number of adaptive samples to $50d$ across all experiments. For the next sample point, we draw $\min(5000d, 50000)$ candidates in each iteration and choose the one that achieves the largest AF

---

[2]https://github.com/SvenL13/LocalValidity

value. The GP is trained in each iteration until 100 samples are observed. Thereafter, training is conducted every 4-th iteration to reduce computational effort. Even if the model is not retrained, we update the GP with the observed data and keep the GP hyperparameters fixed. For the benchmarks, we conduct 30 restarts with different initializations, test sets, and seeds, if not stated otherwise. In every 5-th iteration, we compare the prediction $\tilde{\mathcal{V}}$ (Equation 6) with the GT target based on $\min(25000d, 250000)$ test samples. Additionally, the results for $\tilde{\mathcal{V}}_{0.1}$ are shown in Appendix E.5, and further implementation details of our method can be found in Appendix F.

## 5.1 ANALYTICAL BENCHMARK FUNCTIONS

**Methodology.** For the first benchmark, we consider various analytical functions $f(\mathbf{x})$ to assess method performance under well-controlled conditions, where $f(\mathbf{x})$ represents the error surface ($\delta(\mathbf{x}) := f(\mathbf{x})$). The tested methods have only access to noisy evaluations $f_D(\mathbf{x}) = f(\mathbf{x}) - \epsilon$, where $\epsilon \sim \mathcal{N}(0, \sigma_e^2)$. We use the Styblinsky-Tang [Styblinski and Tang, 1990] and Michalewicz function [Michalewicz, 1992] with varying dimensions, as well as two 2-d benchmark functions popular in RA, namely a modified Rastrigin [Törn and Zhilinskas, 1989] and a series system function [Waarts, 2000] as benchmark functions. Definitions of these analytical functions are given in Appendix G.2.

We compare $\psi_{\mathrm{mis}}$ with $\omega = 0.2\xi$ and $\omega = 0.0$ against several baselines. The first one is given by $\psi_U$ [Echard et al., 2011] adapted to our setting (Equation (4)). Secondly, a random sampling baseline is used, where we select a sample from the candidates with equal probability instead of maximizing an AF. Further, we consider running RA individually for lower and upper tolerance bound with the original U-function [Echard et al., 2011], denoted $\psi_{U2}$. Therefore, we use the sum of two AFs with two GPs, each for $\xi - f_D(\mathbf{x})$ and $\xi + f_D(\mathbf{x})$. Both models observe all samples, even if only one of the GPs is used to query a new sample, i.e., we train and evaluate them on the same inputs. Finally, we consider the smallest margin method [Scheffer et al., 2001] with a XGBoost (XGB) classifier [Chen and Guestrin, 2016], as this seems to be a reasonable baseline [Cawley, 2011, Yang and Loog, 2018] to represent that direction of research. Further settings for the benchmark are given in Table 1.

Table 1: Experimental settings for the analytical benchmark functions.

| Benchmark function | $\xi$ | $\sigma_e$ | $n_{\mathrm{init}}$ | $n_{\mathrm{adapt}}$ |
|---|---|---|---|---|
| Styblinski-Tang | 30 | 5 | 10d | 50d |
| Michalewicz | 0.07 | 0.01 | 10d | 50d |
| Mod. Rastrigin | 20 | 0.1 | 10d | 50d |
| Series System | 3 | 0.5 | 10d | 50d |

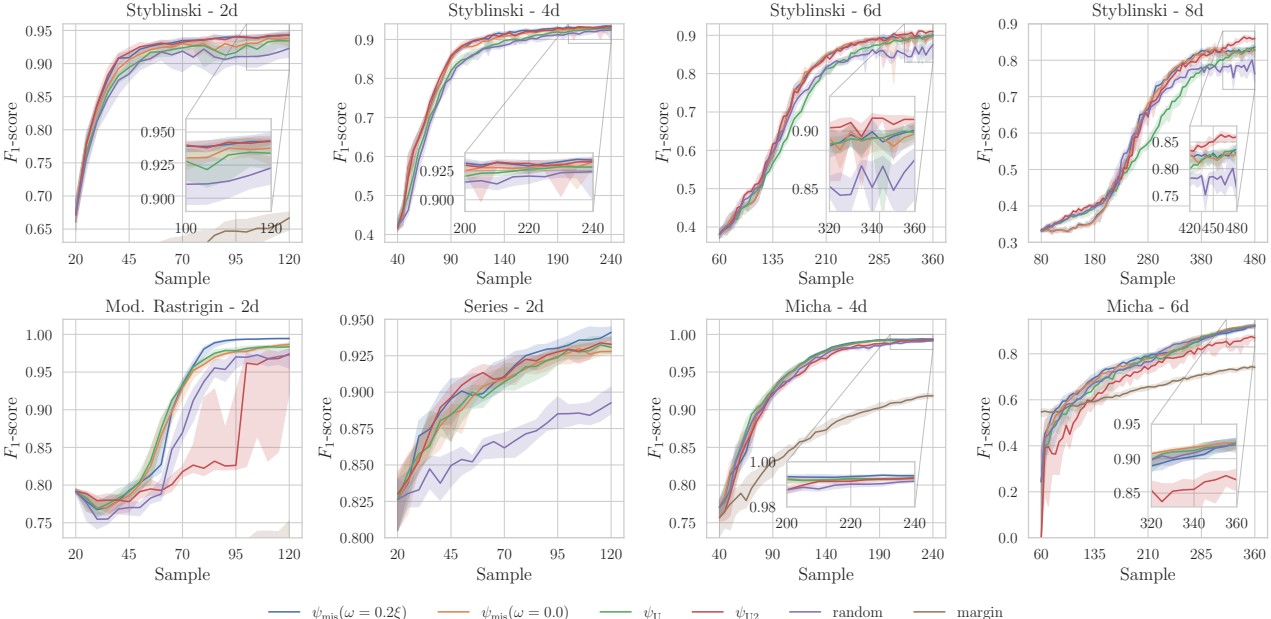

Figure 3: Median and 95% confidence intervals of $F1$-score on the analytical problem functions across 30 runs. Top: Styblinsky-Tang for 2 to 8 dimensions. Bottom: Modified Rastrigin (2-d), series system function (2-d), and Michalewicz function (4-d, 6-d).

**Results.** The results are given in Figure 3, where we report median and 95% confidence bounds. The smallest margin baseline is partially not shown to improve visibility, as it underperforms other methods, with mean $F_1$-score of 39.35% across samples and final score of 73.99% at the last sample. This may be due to the smaller number of available samples compared to applications in previous research. It can be seen that the additional exploration ($\omega = 0.2\xi$) in $\psi_{\mathrm{mis}}$ increases performance slightly upon its counterpart $\psi_{\mathrm{mis}}$ with $\omega = 0$. Further, we found that $\psi_{\mathrm{U2}}$ can be prone to model misspecification, as can be observed for the Rastrigin function, since we have to learn two GPs.

Overall, $\psi_{\mathrm{mis}}$ with $\omega = 0.2\xi$ shows significant improvement over the baselines, achieving an average $F_1$-score of 76.5% and a final score of 92%. In contrast, $\psi_{\mathrm{U2}}$ shows the weakest performance among GP-based strategies, with an average $F_1$-score of 75.21%, slightly improving over the random baseline with 74.55%.

## 5.2 BENCHMARK MODEL VALIDATION

**Methodology.** We evaluate $\psi_{\mathrm{mis}}$ ($\omega = 0.2\xi$), $\psi_{\mathrm{U}}$, and the random baseline for a more realistic problem setting, where the model under validation $f_{\mathrm{M}}$ is given by a trained ML model. Since $\psi_{\mathrm{mis}}(\omega = 0.2\xi)$ outperformed $\psi_{\mathrm{mis}}(\omega = 0)$ on the analytical benchmark functions, we do not expect $\omega = 0$ to perform significantly different here.

Training data for $f_{\mathrm{M}}$ was obtained from an analytical bench-

Table 2: Experimental settings used in ML benchmark.

| Bench. | Dim. | $\xi$ | $\sigma_{\mathrm{e}}$ | $n_{\mathrm{init}}$ | $n_{\mathrm{adapt}}$ |
|--------|------|-------|------------|----------|-----------|
| Micha. | 2 | 0.3 | 0.03 | 20 | 100 |
|        | 4 | 0.6 | 0.03 | 40 | 200 |
|        | 8 | 0.9 | 0.03 | 80 | 400 |
| Rosen. | 2 | 250 | 5 | 20 | 100 |
|        | 4 | 500 | 5 | 40 | 200 |
|        | 8 | 1000 | 5 | 80 | 400 |

mark function $f_{\mathrm{E}}$ via LHS and acquiring noisy labels. In particular, only noisy evaluations of $f_{\mathrm{E}}$ were available both for training and validation, whereas noise-free samples were used for testing the validation models. In all test cases, $f_{\mathrm{M}}$ was ensured to be invalid in some but not all regions (w.r.t input domain). This scenario is common in practice, where a model can perform a task fairly well on average, but remains invalid in certain regions.

Four classes of ML models are considered for $f_{\mathrm{M}}$: RR, SVR, RF, and XGB regression, which exhibit different error surfaces with varying difficulty. $f_{\mathrm{E}}$ is given by the Michalewicz or the Rosenbrock function [Rosenbrock, 1960] with dimensions ranging from 2 to 8. The tolerance $\xi$ is chosen such that we obtain partially valid models, with valid ratio ranging from 0.75 to 0.99. For Rosenbrock we kept hyperparameters of the ML models fixed, while we found the need to tune them via BO for Michalewicz to obtain models that are at least partially valid. Furthermore, we fix the trained

Table 3: Mean and standard error of precision and recall across Ridge regression (RR), support vector regression (SVR), random forest regression (RF), and XGB models. Scores for mean and final result across samples and 30 runs are reported. Bold numbers represent the best result.

| Benchmark | Dimension | Mean Precision [%] | | | Final Precision [%] | | |
|---|---|---|---|---|---|---|---|
| | | $\psi_{\mathrm{mis},0.2}$ | $\psi_{\mathrm{U}}$ | Random | $\psi_{\mathrm{mis},0.2}$ | $\psi_{\mathrm{U}}$ | Random |
| Michalewicz | 2 | $\mathbf{97.3}_{\pm 0.1}$ | $96.9_{\pm 0.1}$ | $96.5_{\pm 0.1}$ | $\mathbf{98.5}_{\pm 0.1}$ | $97.6_{\pm 0.1}$ | $97.4_{\pm 0.1}$ |
| | 4 | $\mathbf{95.1}_{\pm 0.1}$ | $94.9_{\pm 0.1}$ | $94.5_{\pm 0.1}$ | $\mathbf{96.8}_{\pm 0.2}$ | $96.7_{\pm 0.1}$ | $96.2_{\pm 0.2}$ |
| | 8 | $\mathbf{88.5}_{\pm 0.0}$ | $88.4_{\pm 0.0}$ | $88.1_{\pm 0.1}$ | $\mathbf{89.7}_{\pm 0.1}$ | $89.6_{\pm 0.1}$ | $89.2_{\pm 0.1}$ |
| Rosenbrock | 2 | $\mathbf{97.0}_{\pm 0.1}$ | $95.9_{\pm 0.2}$ | $95.5_{\pm 0.1}$ | $\mathbf{98.2}_{\pm 0.1}$ | $97.0_{\pm 0.3}$ | $96.9_{\pm 0.2}$ |
| | 4 | $\mathbf{94.7}_{\pm 0.1}$ | $93.0_{\pm 0.1}$ | $92.1_{\pm 0.1}$ | $\mathbf{96.1}_{\pm 0.1}$ | $94.9_{\pm 0.2}$ | $94.5_{\pm 0.2}$ |
| | 8 | $\mathbf{92.6}_{\pm 0.1}$ | $90.6_{\pm 0.1}$ | $89.9_{\pm 0.1}$ | $\mathbf{94.5}_{\pm 0.1}$ | $92.7_{\pm 0.2}$ | $92.2_{\pm 0.1}$ |
| | | Mean Recall [%] | | | Final Recall [%] | | |
| Michalewicz | 2 | $99.4_{\pm 0.0}$ | $\mathbf{99.6}_{\pm 0.0}$ | $99.1_{\pm 0.1}$ | $99.4_{\pm 0.1}$ | $\mathbf{99.6}_{\pm 0.0}$ | $99.3_{\pm 0.1}$ |
| | 4 | $98.7_{\pm 0.0}$ | $98.8_{\pm 0.0}$ | $\mathbf{99.0}_{\pm 0.1}$ | $98.7_{\pm 0.2}$ | $98.7_{\pm 0.2}$ | $\mathbf{98.9}_{\pm 0.1}$ |
| | 8 | $98.2_{\pm 0.1}$ | $98.3_{\pm 0.1}$ | $\mathbf{98.9}_{\pm 0.1}$ | $97.1_{\pm 0.1}$ | $97.3_{\pm 0.2}$ | $\mathbf{98.0}_{\pm 0.1}$ |
| Rosenbrock | 2 | $98.8_{\pm 0.1}$ | $\mathbf{99.2}_{\pm 0.1}$ | $98.5_{\pm 0.1}$ | $99.2_{\pm 0.1}$ | $\mathbf{99.4}_{\pm 0.1}$ | $98.6_{\pm 0.1}$ |
| | 4 | $93.0_{\pm 0.1}$ | $96.4_{\pm 0.1}$ | $96.2_{\pm 0.2}$ | $95.8_{\pm 0.1}$ | $\mathbf{97.2}_{\pm 0.2}$ | $96.8_{\pm 0.2}$ |
| | 8 | $91.5_{\pm 0.2}$ | $97.0_{\pm 0.1}$ | $\mathbf{97.2}_{\pm 0.1}$ | $92.1_{\pm 0.2}$ | $96.6_{\pm 0.1}$ | $\mathbf{96.8}_{\pm 0.1}$ |

models across the 30 repetitions. The experimental settings are given in Table 2, and further details regarding the ML models hyperparameters and analytical functions can be found in Appendix G.1.

**Results.** The results of our experiments are shown in Table 3, where we report the mean and the last value across samples for precision and recall. We want to emphasize the importance of preventing false positives, i.e., falsely judging a model to be valid, as captured by the precision score from the perspective of safety-critical applications. In contrast, the consequences of false negatives, as reflected by the recall score, are not as severe. Furthermore, we expect the mean across samples to be informative about the overall performance and sample efficiency, while the final value is relevant to the end performance of the method.

It can be seen that $\psi_{\mathrm{mis}}$ outperforms both the baseline and $\psi_{\mathrm{U}}$ in terms of average and final precision across almost all tested cases with regard to precision. While all strategies achieved high recall scores ($> 90\%$), $\psi_{\mathrm{U}}$ and the random sampling baseline slightly outperform $\psi_{\mathrm{mis}}$ in this regard.

## 6  CONCLUSION

Assessing the validity of a model across the range of inputs can be challenging due to the expense of gathering additional validation data. To address this issue, we developed a novel formulation of the local validity problem for ML models inspired by active learning commonly used in RA. Based on this foundation, we proposed a new acquisition function (MC-Prob) that uses the misclassification probability, which can be evaluated in closed-form.

MC-Prob intuitively places samples near the limit state and can learn the boundary between valid and invalid regions of the model. Empirically, MC-Prob improves upon its counterpart, U-Fun, across several benchmarks and two real-world examples, reducing the probability of incorrectly classifying the model as valid, as desired in most safety applications. In contrast to existing conformal prediction methods, our approach can significantly reduce the required amount of data while maintaining accurate predictions. The scalability to higher dimensions remains open, possibly by replacing the GP model [Hensman et al., 2013] or by exploiting the lower intrinsic dimensionality expected in most real-world data [Wang et al., 2016].

### Acknowledgements

This work was supported by ZF Friedrichshafen AG.

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

# Quantifying Local Model Validity via Active Learning
# (Supplementary Material)

**Sven Lämmle**[1, 2]  **Can Bogoclu**[3]  **Robert Voßhall**[4]  **Anselm Haselhoff**[5]  **Dirk Roos**[2]

[1]Center of Expertise, ZF Friedrichshafen AG, Friedrichshafen, Germany
[2]Institute of Modelling and High-Performance Computing, Niederrhein University of Applied Sciences, Krefeld, Germany
[3]Zalando SE, Berlin, Germany
[4]auxmoney GmbH, Düsseldorf, Germany
[5]Ruhr West University of Applied Sciences, Bottrop, Germany

## A  DERIVATION OF EQUATION 5

Let $Y = |X|$, where $X \sim \mathcal{N}(\mu, \sigma^2)$ with mean $\mu$ and variance $\sigma^2$. Then, $Y$ follows a folded Gaussian [Leone et al., 1961], with parameters $\mu$ and $\sigma^2$. The cdf is given by [Tsagris et al., 2014]

$$F(x) = 0.5 \left( erf\left(\frac{x-\mu}{\sigma\sqrt{2}}\right) + erf\left(\frac{x+\mu}{\sigma\sqrt{2}}\right) \right),$$

where $erf(x) = 2/\sqrt{\pi} \int_0^x \exp(-t^2)dt$ is the error function. Further, let $Z = a - Y = a - |\mu + \sigma X|$. Then, for $x \in (0, \infty)$,

$$F(x) = P(Z \le x) = P(a - |\mu + \sigma X| \le x) = 1 - P(|\mu + \sigma X| \le \underbrace{-x + a}_{:=z}) = 1 - P(-z \le \mu + \sigma X \le z)$$

$$= 1 - P\left(\frac{-z-\mu}{\sigma} \le X \le \frac{z-\mu}{\sigma}\right)$$

$$= 1 + \Phi\left(\frac{-z-\mu}{\sigma}\right) - \Phi\left(\frac{z-\mu}{\sigma}\right)$$

$$= 2 - \Phi\left(\frac{z+\mu}{\sigma}\right) - \Phi\left(\frac{z-\mu}{\sigma}\right),$$

where $\Phi(\cdot)$ is the standard normal cdf, and $z := -x + a$. The last step follows since $\Phi(-x) = 1 - \Phi(x)$. We obtain the formulation in Equation 5 by using $x = \omega$ and $a = \xi$, with folded Gaussian parameters $\mu$ and $\sigma^2$ given by the GP predictive mean and variance. See Figure 4 for a illustration.

## B  LIMITATIONS AND DISCUSSION

We have shown that our approach can learn the limit state $\mathcal{S}$ and predict the valid set $\mathcal{V}$ with a reduced number of validation samples. However, it is important to discuss further challenges and limitations we encountered during development.

Firstly, we observed that high label noise in relation to the tolerance level $\xi$ can degrade the performance of the strategy. In such cases, the underlying limit state may not be accurately identifiable, as shown by our ablation study in Appendix E.2. Therefore, if small $\xi$ has to be achieved, it is important to keep the noise correspondingly low.

Secondly, misspecification of the GP model is important to consider and could occur, e.g., if many hyperparameters have to be optimized. To mitigate possible misspecification, we train the GP model multiple times with different parameter initializations (see implementation details in Appendix F). Additionally, one can provide suitable prior distributions for the model hyperparameters if available. Issues could arise with discontinuous or unsmooth error surfaces, as observed during validation of tree-based models, due to the learned decision structure. In such situations, the GP model can only provide a smooth approximation.

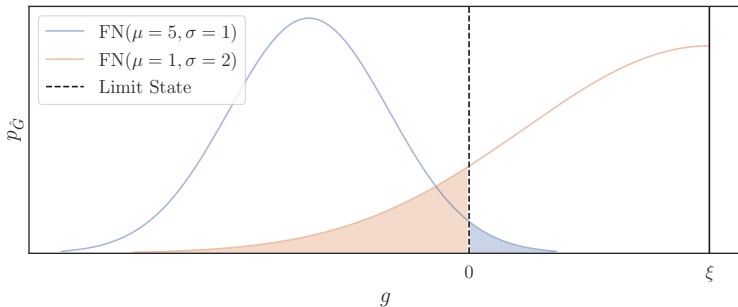

Figure 4: GP prediction of the limit state function $g$ is a folded Gaussian distribution, which is flipped and shifted by the predefined tolerance $\xi$. The filled area shows the misclassification probability $\psi_{\mathrm{mis}}$.

Finally, our method was developed in the setting of additive homoscedastic Gaussian noise with variance $\sigma_{\mathrm{e}}^2$. However, real-world applications can be influenced by heteroscedastic noise, where $\sigma_{\mathrm{e}}^2$ may change with $\mathbf{x}$. Thus, the estimated noise would be under- or overestimated in certain input regions if assumed homoscedastic. Nevertheless, homoscedasticity can be a reasonable assumption even for real-world data. This is demonstrated in Appendix E.1 by applying our method with homoscedasticity assumption to real-world examples. To further improve the performance with heteroscedastic noise, a suitable transformation on the labels could be applied, such as the Box-Cox [Box and Cox, 1964] or Yeo-Jonhson transformation [Yeo and Johnson, 2000], which have been used in the context of BO and GP models [Cowen-Rivers et al., 2022]. Alternatively, the noise can be learned directly by a second GP model [Kersting et al., 2007, Binois et al., 2018]. Testing such an approach is left for future work.

## C    ADDITIONAL THEORETICAL CONSIDERATIONS

### C.1    ERROR BOUNDS

In the following, we discuss the error bound using the 90% confidence interval of the GP model and the error bounds based on Theorem 1. Therefore, a 1-d test function

$$\delta(x) = \frac{1}{2} \, \exp(x) \, \sin(8x - 2)$$

and additive noise ($\sigma_{\mathrm{e}}^2 = 0.05^2$) is used with Algorithm 1. Tolerance is set to $\xi = 1$, and we use 10 initial samples and 500 iterations. The two limit states are at $x_1 \approx 0.786$ and $x_2 \approx 0.92$. For this case, it is possible to compute $\eta$ given in Theorem 1 with exact Lipschitz constants.

The results are illustrated in Figure 5. It can be seen, that the bound using the 90% confidence interval of the GP model may underestimate the true error, but yields in most cases a good bound. In contrast, the error bound given by $\eta$ is in any case a very conservative approximation of the true error. Further, it can be seen that our adaptive sampling strategy results in a rapid decrease of the error bound $\eta$ near the limit state even for small sample sizes, whereas the model improves globally with more samples available.

Figure 6 shows the comparison of $\eta$ with exact Lipschitz constants and $\eta$ calculated using the bounds from Equation 8 and Equation 9. The setting is the same as in the top left of Figure 5. Apparently, the upper bounds for the Lipschitz constant yield too pessimistic results for small sample sizes.

Theorem 1 states that the true error is less than the error bound $\eta$ for all points in the input space with a probability of 90%. Therefore, $\eta$ gives a uniform error bound that is not directly dependent on the accuracy of the predicted uncertainty of the GP, and is therefore much stronger. However, the theorem requires additional knowledge on the Lipschitz continuity of the covariance kernel as well as $\delta$ and is therefore not generally applicable. Furthermore, Figures 5, 6 show that the bound may be too conservative, especially when using estimated Lipschitz constants.

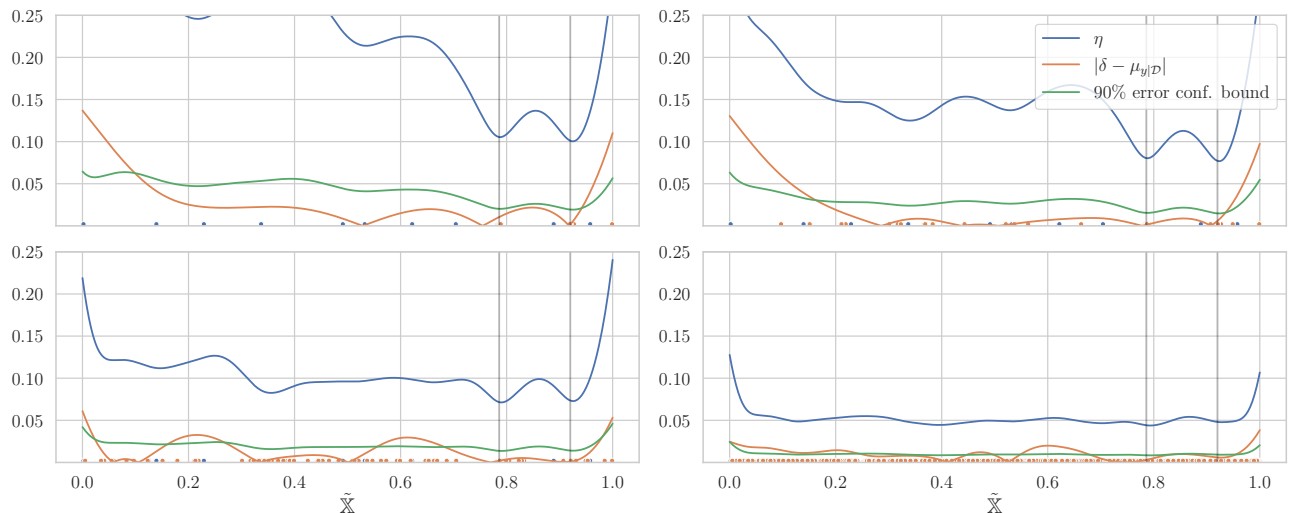

Figure 5: Error bound $\eta$ for $|\delta - \mu_{y|\mathcal{D}}|$ as well as true error and 90% confidence interval of the GP model for 10 initial samples, 20 (top left), 50 (top right), 100 (lower left) and 500 (lower right) adaptive samples. The vertical lines show the limit states. Initial and adaptive samples are blue and orange, resp.

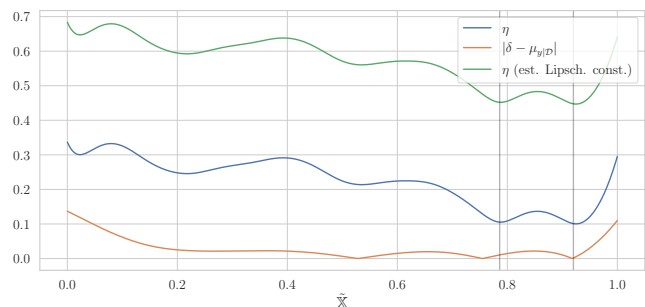

Figure 6: Error bound $\eta$ for $|\delta - \mu_{y|\mathcal{D}}|$ with exact Lipschitz constants and using the bounds given in (Equation 8) and (Equation 9).

## C.2 PROBABILITY OF MISCLASSIFICATION

The probability of misclassification at $\mathbf{x}$ is given by Equation 5 and derived in Appendix A. For the case $\mu < -\xi$ (with $\mu = -4$ and $\xi = 2$), this probability is illustrated as a yellow area in Figure 7. This setting corresponds to the case of an invalid posterior mean. The misclassification probability is the probability that the state is actually valid. Since the valid domain is bounded, this probability can only be maximized up to some limited extent and it is bounded by 0.5 in any case. In contrast, the misclassification probability of a state that is predicted to be valid can be arbitrarily close to one and is strictly increasing with the variance for fixed $\mu$ (see right-hand side of Figure 7). For fixed $\mu$ it is even possible to compute the $\sigma$ which maximizes the probability of misclassification. This can be achieved by taking the derivative of Equation 5 with respect to $\sigma$ and by computing its root. The optimal standard deviation $\sigma_{\text{opt}}$ is given by

$$\sigma_{\text{opt}}^2 = -2\,\xi\,\mu\,\ln\left(\frac{\mu - \xi}{\mu + \xi}\right)^{-1} \quad \text{for fixed } \mu > |\xi|.$$

Figure 8 shows the dependence of the misclassification probability on both $\mu$ and $\sigma$. The optimal standard deviation increases with the distance of $\mu$ to $\xi$ and very small variance results in small values for $P_{\text{mis}}$. However, as described before, $P_{\text{mis}}$ decreases again if $\sigma$ becomes too large.

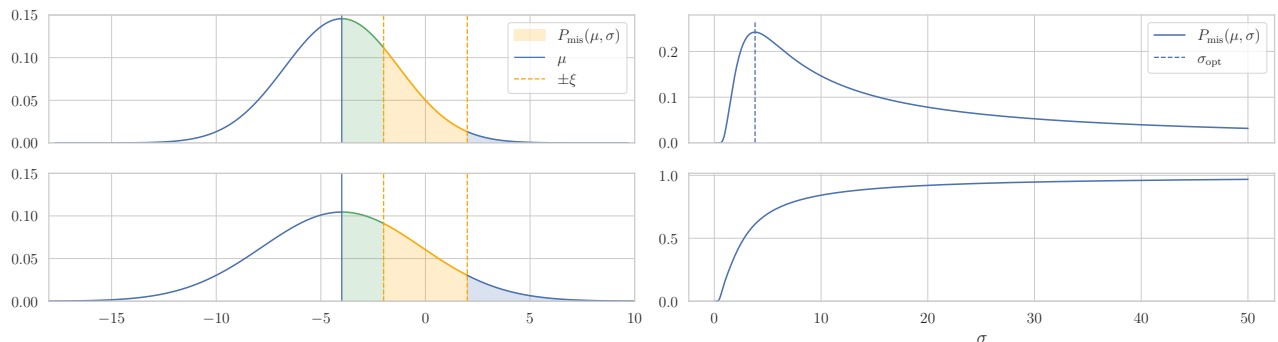

Figure 7: Left: Yellow area equals the misclassification probability and is maximized for $|\mu| > \xi$ if green and blue area are equal (lower figure). Right: Misclassification probability for fixed $\mu$ if $|\mu| > \xi$ (upper figure) and $|\mu| \le \xi$ (lower figure).

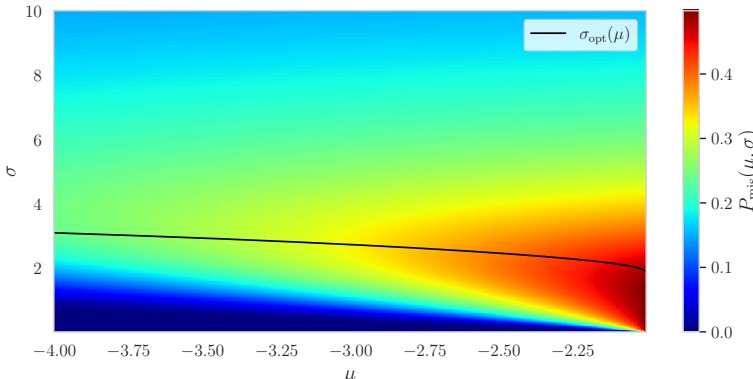

Figure 8: Dependence of the misclassification probability on $\mu$ and $\sigma$ for $\mu < -\xi$ with $\xi = 2$.

## D    COMPARISON WITH CONFORMAL PREDICTION

While our primary objective of this work is designing an adaptive sampling strategy for learning the limit state $\mathcal{S}$ (Section 4), an important question arises how our approach compares with existing conformal prediction methods. Hence, we provide a comparison with the most popular conformal prediction strategies, which we can utilize to predict the valid set $\tilde{\mathcal{V}}_\alpha$ for the model under validation $f_{\mathrm{M}}$.

**Prediction Interval and Valid Set.**    Conformal prediction methods are used to derive prediction intervals $\hat{C}_\alpha(\mathbf{x})$ that contain an unseen observation $Y^\star$ at test point $X^\star$ with confidence $1 - \alpha$ (often referred as coverage), where no further assumption is made about the data generating distribution $p(\mathbf{x}, y)$. The framework provides guarantees, with the most common one being *marginal coverage*, which aims to satisfy

$$P(Y^\star \in \hat{C}_\alpha(X^\star)) \ge 1 - \alpha.$$

Note, in order to form a prediction interval, most strategies (e.g., split conformal prediction) use additional calibration data besides the training data for $f_{\mathrm{M}}$ [Lei et al., 2018, Bellotti, 2020]. In our setting, we use the validation data for this purpose.

We can obtain a valid set similar to $\mathcal{V}_\alpha$ (Equation 7), by using lower $\rho_{\mathrm{lo}}$ and upper bound $\rho_{\mathrm{lo}}$ of the prediction interval, where $\hat{C}_\alpha(\mathbf{x}) = [\rho_{\mathrm{lo}}(\mathbf{x}), \rho_{\mathrm{up}}(\mathbf{x})]$. Further, we can obtain the valid set from $\hat{C}_\alpha(\mathbf{x})$ as

$$\tilde{\mathcal{V}}_\alpha = \{\mathbf{x} \in \mathbb{X} : (f_{\mathrm{M}}(\mathbf{x}) - \rho_{\mathrm{lo}}(\mathbf{x}) \le \xi) \wedge (\rho_{\mathrm{up}}(\mathbf{x}) - f_{\mathrm{M}}(\mathbf{x}) \le \xi)\},$$

where $\xi \in \mathbb{R}_+$ is the tolerance level. In other words, we classify $f_{\mathrm{M}}$ to be valid at $\mathbf{x}$ if we predict with $1 - \alpha$ confidence that the difference between mean prediction ($f_{\mathrm{M}}$) and upper/lower bound is below the prescribed tolerance $\xi$. Thus, we are able to compare conformal prediction methods to our method based on $\tilde{\mathcal{V}}_\alpha$.

Table 4: Qualitative comparison between strategies to perform validation. Split conformal and MAD have only marginally coverage, while LVD provides approximately conditional coverage. Similarly, our method can be seen to have marginal coverage if the error bound from Theorem 1 is used. In our experiments, conformal methods have shown the need for larger datasets in order to provide meaningful prediction intervals for validation, in comparison to the proposed approach (Section 4).

| | Split Conformal | MAD | LVD | Ours |
|---|---|---|---|---|
| Coverage | marginal | marginal | apprx. conditional | marginal[1] |
| Discriminative | ✗ | ✓ | ✓ | ✓ |
| Applicable $f_M$ | regression, class. | regression | regression | regression |
| Samples needed | medium | high | high | small |

[1]based on Theorem 1

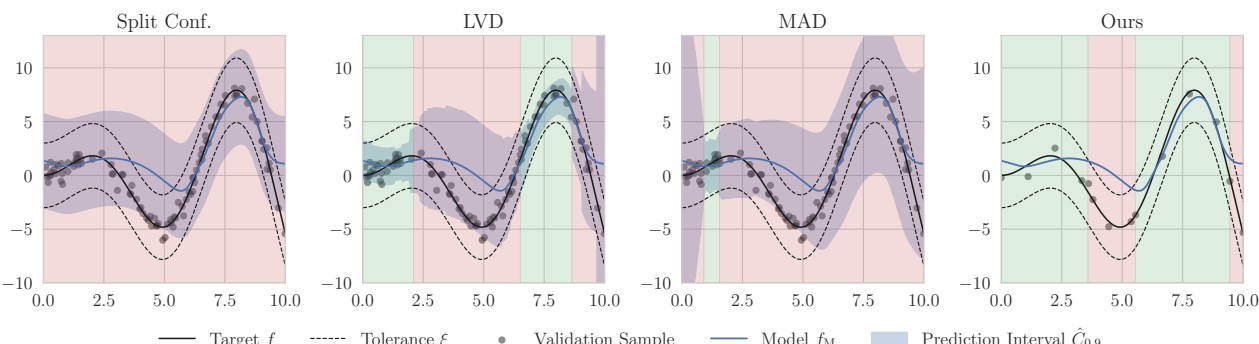

Figure 9: Comparison of different validation strategies, where we used 100 validation samples for the conformal strategies (Split conformal, LVD and MAD), and 14 samples for our method. The target is to obtain an accurate estimate of the valid regions, i.e., $f_M$ is inside the tolerance $\xi$. The prediction of the local valid set $\tilde{\mathcal{V}}_{0.1}$ (●) is shown for each method.

## D.1 COMPARISON

We compare our approach (Section 4) with the following methods from conformal prediction:

- Split-conformal prediction [Papadopoulos et al., 2002], with the commonly-used residual score $s(\mathbf{x}, y) = |y - f_M(\mathbf{x})|$.
- Mean absolute deviation - normalized split conformal (MAD) [Lei et al., 2018, Bellotti, 2020], with the residual score $s(\mathbf{x}, y) = |y - f_M(\mathbf{x})|/u(\mathbf{x})$, where $u(x)$ is the residual predictor for $|y - f_M(\mathbf{x})|$.
- Locally valid discriminative prediction intervals (LVD) [Lin et al., 2021], with squared exponential kernel.

A qualitative comparison between methods is given in Table 4.

**Simple Example.** To illustrate differences in the obtained valid sets, we train a GP model $f_M$ to be validated on 10 samples from $f(x) = x \sin(x) + \epsilon$, with $\epsilon \sim \mathcal{N}(0, 0.5^2)$ (similar to Figure 1). $x$ is drawn uniformly between 0 and 10, and tolerance $\xi$ is set to 3. To obtain the valid set for the conformal methods, a total of 100 samples are drawn uniformly. For our method, we use 10 initial samples and 4 samples drawn with $\psi(\omega = 0.1\xi)$. A confidence level of $1 - \alpha = 0.9$ is used for all methods, while predictions are given by $\tilde{\mathcal{V}}_{0.1}$. Results are shown in Figure 9. We see that the prediction interval with split conformal has a fixed width for all $\mathbf{x}$, since the method is not discriminative. Therefore, we see the method is not capable of identifying valid regions in this setting. LVD is discriminative and therefore able to correctly identify some parts of the valid region. However, with few data, the prediction intervals of LVD can become infinitely wide, as noted by the authors [Lin et al., 2021, Section 3.2]. For MAD, only a small region is correctly classified as valid. In contrast, our method is able to identify valid regions with high accuracy. Further, we found that all tested conformal strategies need considerably more samples than our method in order to provide meaningful prediction intervals.

**8-dimensional Comparison.** We compare methods based on the 8-dimensional benchmark from Section 5.2, where we used RR, SVR, RF, and XGB as $f_M$ to be validated. For comparison, we use the final result obtained from our method, with

Table 5: Comparison of different validation strategies for the 8-dimensional ML benchmark (Section 5) with RR, SVR, RF, and XGB models. Mean and standard error are shown across 10 runs for the conformal prediction strategies (Split Conf., MAD and LVD) and 30 runs for our strategy (Section 4). Predictions are made with $\tilde{\mathcal{V}}_{0.1}$. Scores for the final sample are reported for our method. Bold numbers represent the best result.

| Benchmark | Metric | Method | | | |
| --- | --- | --- | --- | --- | --- |
| | | Split Conf. | MAD | LVD | Ours |
| Michalewicz (8-d) | $F_1$-score [%] | $24.9_{\pm 6.9}$ | $44.4_{\pm 4.1}$ | $0.0_{\pm 0.0}$ | $\mathbf{74.5}_{\pm 1.8}$ |
| | Precision [%] | $24.9_{\pm 6.9}$ | $90.2_{\pm 0.9}$ | $0.0_{\pm 0.0}$ | $\mathbf{94.0}_{\pm 0.4}$ |
| | Recall [%] | $25.0_{\pm 6.9}$ | $33.4_{\pm 4.3}$ | $0.0_{\pm 0.0}$ | $\mathbf{64.8}_{\pm 2.1}$ |
| Rosenbrock (8-d) | $F_1$-score [%] | $24.8_{\pm 6.9}$ | $52.4_{\pm 4.8}$ | $0.0_{\pm 0.0}$ | $\mathbf{64.9}_{\pm 2.4}$ |
| | Precision [%] | $24.6_{\pm 6.8}$ | $93.0_{\pm 0.6}$ | $0.0_{\pm 0.0}$ | $\mathbf{98.0}_{\pm 0.2}$ |
| | Recall [%] | $25.0_{\pm 6.9}$ | $43.2_{\pm 5.5}$ | $0.0_{\pm 0.0}$ | $\mathbf{53.3}_{\pm 2.5}$ |

$\psi_{\mathrm{mis}}$ and $\omega = 0.2\xi$. Further, we draw uniformly the same number of samples as used by our adaptive approach (480 samples for 8-d) to calibrate the prediction intervals. We show the average across 10 runs for the conformal prediction strategies and 30 runs with our method (results taken from Table 6) for $F_1$-score, precision, and recall. In Table 5, it can be seen that our method outperforms the conformal strategies by a large margin across all metrics. Across the conformal methods, MAD provided the best performance. We found that LVD had issues with the small dataset, leading to infinitely wide prediction intervals in most cases, as described previously.

## D.2   CONCLUSION

Methods for conformal prediction were derived to provide prediction intervals without further assumption of the underlying distribution. In our setting, we found that the resulting prediction intervals were overly conservative when used for validation, especially with limited data. Furthermore, LVD provided infinitely wide prediction intervals if not enough samples are available, which makes the method difficult to use if data is scarce. In contrast, we have seen that our proposed method can provide accurate valid sets without the need for excessive amounts of samples. The usefulness of conformal prediction strategies is their wide applicability, where they can be used in the more general setting with heteroscedastic noise, i.e., $\sigma_{\mathrm{e}}^2$ may change with $\mathbf{x}$. Future research may extend our proposed approach, since GP models are well capable of handling heteroscedastic noise, as shown with other active learning strategies [Binois et al., 2018, 2019].

## E   ADDITIONAL RESULTS

Here, we present additional experimental results complementing the benchmarks shown in Section 5.

### E.1   TABULAR DATASET

We show the practical application of our method by extending experiments from Section 5.2 with two real-world tabular datasets (6-d: ID-4835, 9-d: ID-361083) from `OpenML` [Vanschoren et al., 2013], containing numerical features. In contrast to previous experiments, the available data is restricted, i.e., we cannot place adaptive samples at arbitrary $\mathbf{x} \in \tilde{\mathbb{X}}$ (previously we could query $f_{\mathrm{E}}$ which should be observable in practice). Hence, the available data is randomly separated into train $\mathcal{D}_{\mathrm{train}}$ (6-d: 621 samples, 9-d: 349100 samples), validation $\mathcal{D}_{\mathrm{val}}$ (6-d: 1234 samples, 9-d: 93093 samples), and test sets $\mathcal{D}_{\mathrm{test}}$ (6-d: 1234 samples, 9-d: 139641 samples). Furthermore, if the active learning strategy proposes to query at a position $\mathbf{x}^\star$ we pick the closest ($L^2$ distance) available sample $\mathbf{x}' \in \mathcal{D}_{\mathrm{val}}$ in the validation set and return the corresponding observation $y' \in \mathcal{D}_{\mathrm{val}}$. We compare $\psi_{\mathrm{mis},0.2}$, $\psi_{\mathrm{U}}$, and the random sampling baseline.

**Results 6-dimensional dataset.**   Two models have to be validated: a GP model (Matérn5/2 kernel) and a SVR model with default hyperparameters. The models were trained with $\mathcal{D}_{\mathrm{train}}$, which results in a valid ratio of 0.64 for the GP and 0.65 for the SVR model, with $\xi = 0.1$. Figure 10 shows median and 95% confidence interval for the $F1$-score across 20 runs. The dashed lines represent a reference GP model trained with the complete validation data $\mathcal{D}_{\mathrm{val}}$.

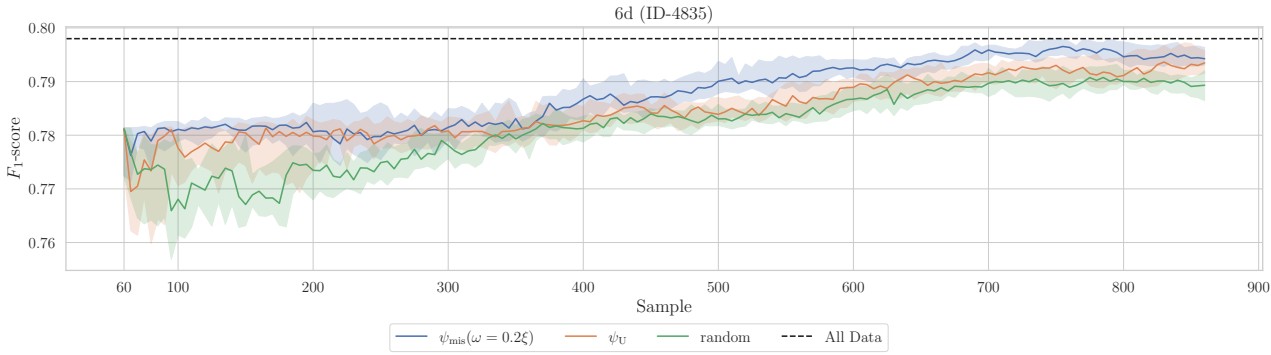

Figure 10: Median and 95% confidence intervals of $F1$-score on the 6-dimensional tabular dataset for GP and SVR model across 20 runs. Dashed line represents reference model with all available samples from the tabular dataset.

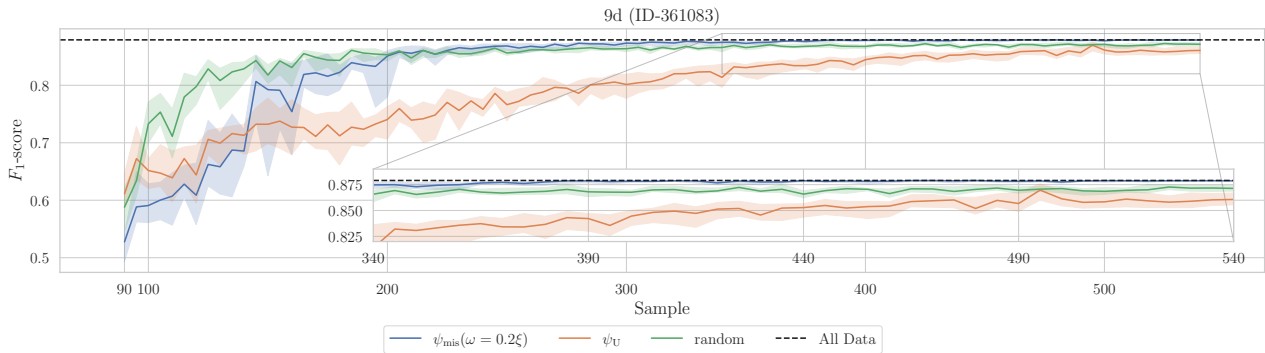

Figure 11: Median and 95% confidence intervals of $F1$-score on the 9-dimensional tabular dataset for XGB model and across 30 runs. Dashed line represents reference model with all available samples from the tabular dataset.

**Results 9-dimensional dataset.** The model to be validated is a XGB model trained on $\mathcal{D}_{\text{train}}$ with default hyperparameters, with test $R^2 \approx 0.4$ and tolerance $\xi = 0.4$, which gives a valid ratio of 0.78. Due to the computational burden, a sparse variational Gaussian process regression (SVGP) model [Hensman et al., 2013, Titsias, 2009] is used instead of the GP (Section 4.3). See Appendix F for the SVGP implementation details.

Figure 11 shows median and 95% confidence interval for the $F1$-score across 30 runs. Further, we show a reference SVGP model (dashed lines) with the complete validation data $\mathcal{D}_{\text{val}}$ (11636 samples). It can be seen that although random sampling improves the score faster in the beginning, only $\psi_{\text{mis}}$ achieves a score close to the reference solution.

### E.2 INFLUENCE OF NOISE ON MODEL QUALITY

Noise $\sigma_{\text{e}}^2$ can have a non-neglectable impact on the achievable model accuracy and can lead, if high enough, to identifiability issues of the underlying limit state. Figure 12 shows the influence of varying noise on the $F_1$-score and the predicted misclassification probability $\tilde{P}_{\text{mis}}$ for the 2-dimensional Styblinsky-Tang function. In this experiment, we varied the signal-to-noise ratio N/S between 0.1% and 50%. The ratio is calculated as N/S $= \sigma_{\text{e}}^2 / \mathbb{E}_{\boldsymbol{X} \sim p(\mathbf{x})} \left[ f(\mathbf{X}) \right]^2$, where $f(\cdot)$ is the Styblinsky-Tang function, and $p(\mathbf{x})$ is taken to be uniform. The expectation is numerically evaluated. Furthermore, we provide the signal to tolerance ratio (N/T $= \sigma_{\text{e}} / \xi$). It can be observed, that in the most extreme case (N/S $= 50\%$), the final $F_1$-score has decreased on average from 0.95 to 0.58. Note, that this performance decrease is captured by $\tilde{P}_{\text{mis}}$.

### E.3 INFLUENCE OF TOLERANCE LEVEL

The chosen tolerance level $\xi$ can influence the efficacy of the adaptive strategy and the learned error model $\hat{f}_{\text{D}}$. A stricter tolerance will most likely result in more invalid regions, making the problem more challenging to model, except in cases

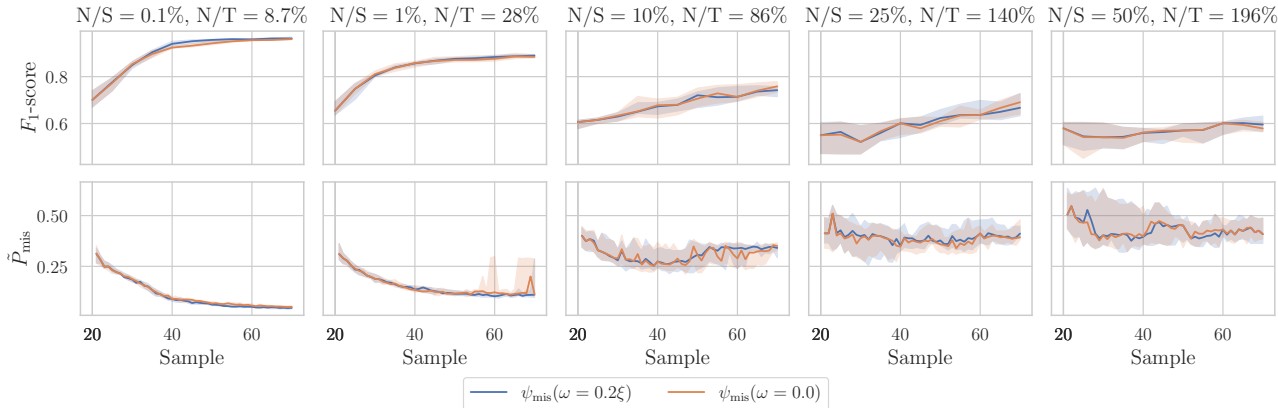

Figure 12: Varying noise to signal (N/S $= \sigma_e^2 / \mathbb{E}_{\boldsymbol{X} \sim p(\mathbf{x})} \left[ f(\mathbf{X}) \right]^2$) or noise to tolerance (N/T $= \sigma_e / \xi$) ratios across columns. Median and 95% confidence intervals for the 2-dimensional Styblinsky-Tang function (valid to invalid ratio: 0.77) after 20 initial observations and across 25 runs are shown.

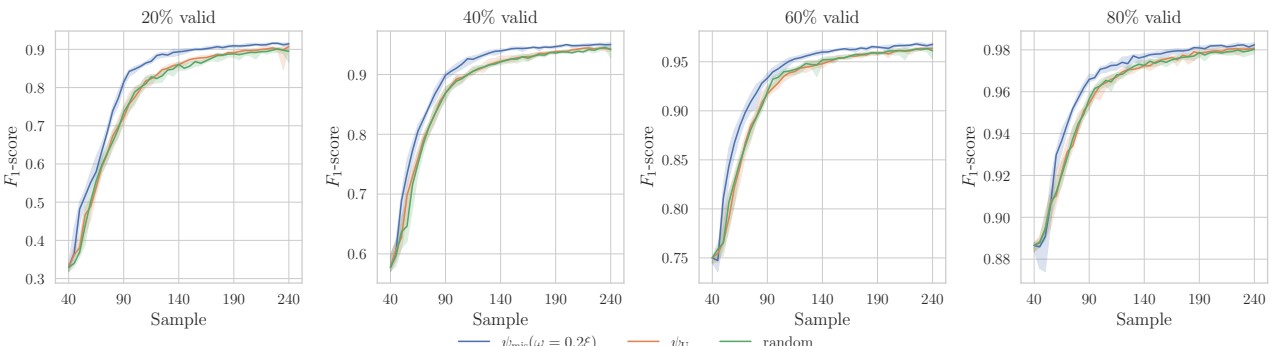

Figure 13: Varying tolerance levels $\xi$ for the 4-d Styblinsky-Tang function, selected to obtain four valid ratios. Median and 95% confidence intervals are shown.

where everything is invalid, which makes the problem much easier. Note that this is a property inherent to the problem. Figure 13 shows the results for the 4-d Styblinsky-Tang function with different tolerance levels, selected to obtain four valid ratios (20%, 40%, 60%, and 80%) for $f_M$. Other settings were kept the same as in Section 5.

### E.4 STOPPING CRITERION

We show the results for the misclassification probability $\tilde{P}_{\mathrm{mis}}$, proposed in Section 4.5 as a stopping criterion, which we tracked during experiments in Section 5.2. From this, we evaluate the difference between the misclassification rate calculated from the test data and our stopping criterion $\tilde{P}_{\mathrm{mis}}$ based on the trained GP models. Results are shown averaged across 2, 4, and 8 dimensions in Figure 14. It can be seen that for both AFs, $\tilde{P}_{\mathrm{mis}}$ tends to be slightly conservative. Similar observations were made for the original U-function, see [Wang and Shafieezadeh, 2019]. Nevertheless, the stopping criterion provides to be useful in combination with a maximum sample budget, as it can lead to early stopping if a sufficient number of samples are obtained.

### E.5 LOWER RISK AVERSITY

Complementary results for the benchmark in Section 5.2 using $\tilde{\mathcal{V}}_{0.1}$ (Equation 7) are given in Table 6. We observe that $\psi_{\mathrm{mis}}$ shows overall the highest precision score on average and final score, outperforming $\psi_U$ and the random baseline. Due to the more conservative predictions, there is a drop in the recall score for all methods. Differences between the random sampling baseline and the AFs can be explained by the limited exploration of the AFs in comparison.

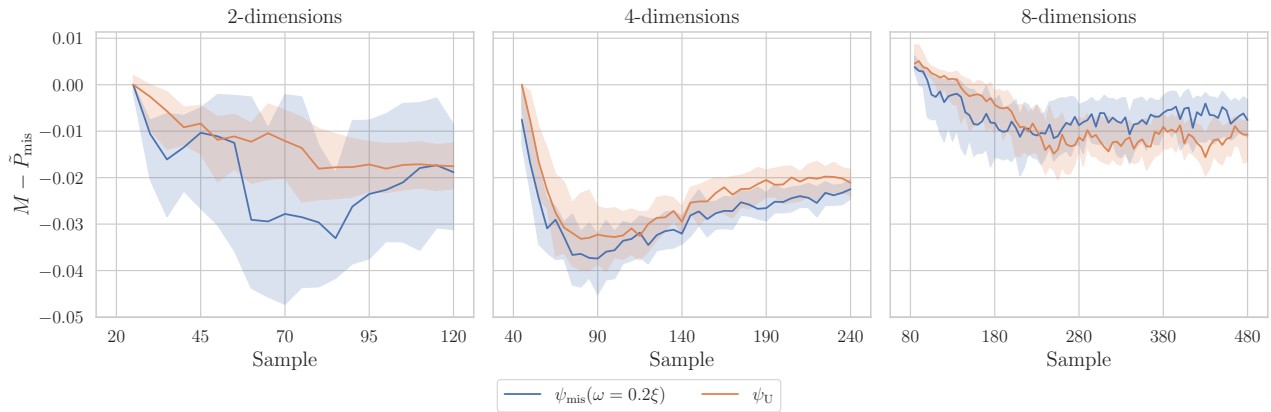

Figure 14: Difference between true misclassification rate $(M)$ and $\tilde{P}_{\text{mis}}$ across RR, SVR, RF, and XGB. Median and 95% confidence intervals are shown.

Table 6: Mean and standard error for precision and recall across 30 runs for RR, SVR, RF, and XGB. Predictions are made with $\tilde{\mathcal{V}}_{0.1}$. Scores for mean and maximum across samples are reported. Bold numbers represent the best result.

| Benchmark | Dimension | Mean Precision | | | Final Precision | | |
|---|---|---|---|---|---|---|---|
| | | $\psi_{\text{mis},0.2}$ | $\psi_{\text{U}}$ | Random | $\psi_{\text{mis},0.2}$ | $\psi_{\text{U}}$ | Random |
| Michalewicz | 2 | $\mathbf{98.5}_{\pm 0.1}$ | $98.0_{\pm 0.2}$ | $98.2_{\pm 0.1}$ | $\mathbf{99.0}_{\pm 0.2}$ | $98.4_{\pm 0.2}$ | $98.8_{\pm 0.1}$ |
| | 4 | $\mathbf{98.0}_{\pm 0.1}$ | $97.8_{\pm 0.1}$ | $97.6_{\pm 0.1}$ | $\mathbf{99.1}_{\pm 0.1}$ | $99.0_{\pm 0.1}$ | $98.9_{\pm 0.1}$ |
| | 8 | $\mathbf{91.9}_{\pm 0.1}$ | $91.7_{\pm 0.2}$ | $91.6_{\pm 0.2}$ | $\mathbf{94.0}_{\pm 0.2}$ | $93.9_{\pm 0.3}$ | $93.9_{\pm 0.3}$ |
| Rosenbrock | 2 | $\mathbf{98.5}_{\pm 0.1}$ | $97.8_{\pm 0.2}$ | $98.0_{\pm 0.2}$ | $\mathbf{98.9}_{\pm 0.1}$ | $98.3_{\pm 0.3}$ | $98.8_{\pm 0.1}$ |
| | 4 | $\mathbf{98.1}_{\pm 0.1}$ | $97.8_{\pm 0.1}$ | $97.7_{\pm 0.1}$ | $98.1_{\pm 0.2}$ | $98.2_{\pm 0.1}$ | $\mathbf{98.3}_{\pm 0.1}$ |
| | 8 | $\mathbf{97.2}_{\pm 0.1}$ | $96.5_{\pm 0.1}$ | $95.7_{\pm 0.1}$ | $\mathbf{98.0}_{\pm 0.1}$ | $97.6_{\pm 0.1}$ | $97.2_{\pm 0.1}$ |
| | | Mean Recall | | | Final Recall | | |
| Michalewicz | 2 | $69.3_{\pm 1.8}$ | $72.1_{\pm 2.1}$ | $\mathbf{88.8}_{\pm 0.9}$ | $75.4_{\pm 1.0}$ | $75.5_{\pm 2.7}$ | $\mathbf{93.1}_{\pm 1.1}$ |
| | 4 | $73.2_{\pm 0.7}$ | $72.2_{\pm 0.7}$ | $\mathbf{77.7}_{\pm 0.6}$ | $82.6_{\pm 1.7}$ | $81.1_{\pm 2.1}$ | $\mathbf{85.4}_{\pm 1.1}$ |
| | 8 | $\mathbf{51.3}_{\pm 1.3}$ | $42.0_{\pm 1.1}$ | $50.3_{\pm 0.9}$ | $\mathbf{64.8}_{\pm 2.1}$ | $52.4_{\pm 2.4}$ | $63.3_{\pm 1.5}$ |
| Rosenbrock | 2 | $66.1_{\pm 1.4}$ | $67.2_{\pm 2.1}$ | $\mathbf{88.0}_{\pm 1.2}$ | $68.1_{\pm 1.6}$ | $66.1_{\pm 2.9}$ | $\mathbf{93.2}_{\pm 0.5}$ |
| | 4 | $44.4_{\pm 1.0}$ | $49.7_{\pm 1.0}$ | $\mathbf{65.1}_{\pm 1.3}$ | $55.4_{\pm 2.7}$ | $57.1_{\pm 3.4}$ | $\mathbf{75.4}_{\pm 2.2}$ |
| | 8 | $38.1_{\pm 0.7}$ | $48.8_{\pm 0.9}$ | $\mathbf{63.5}_{\pm 0.8}$ | $53.3_{\pm 2.3}$ | $61.5_{\pm 2.8}$ | $\mathbf{73.7}_{\pm 1.6}$ |

# F  IMPLEMENTATION DETAILS

We now provide further implementation details for our method.

**Sampling Strategy.** For generating initial and candidate samples, we used a quasi-Monte Carlo sampling strategy based on LHS. The method is implemented in [Bogoclu et al., 2021], where samples are drawn from $\mathbb{X}$ without correlation and by maximizing pairwise distance. As an alternative, one could use `SciPy`'s [Virtanen et al., 2020] quasi-Monte Carlo implementations (e.g., LHS or Sobol sampling). We draw a new set of candidates in each iteration of the adaptive method.

**Gaussian Process Model.** The GP model used throughout this work is implemented with `GPyTorch` [Gardner et al., 2018] and on top of `BoTorch` [Balandat et al., 2020]. Furthermore, we transformed the inputs of the model to the unit cube. Output of the model was normalized to be zero mean and unit variance during training, and reversed for prediction. The GP uses a sum of five kernels: squared exponential, Matèrn 1/2, Matèrn 3/2, Matèrn 5/2, and rational quadratic, where we placed half-Cauchy priors ($\sigma = 2$) on the lengthscales. The reason for using this combination is: **1)** the five kernels provide significant flexibility. **2)** The heavy tales of the half-Cauchy prior can disable unused dimensions according to the principle of *automatic relevance determination* [MacKay and Neal, 1994]. **3)** Our preliminary tests showed that using this combination provides improved performance, e.g., in contrast to using the Matèrn 3/2, although being more difficult to train.

Table 7: BO search space for hyperparameter optimization.

| RR | | SVR | | RF | | XGB | |
|---|---|---|---|---|---|---|---|
| Parameter | Distribution | Parameter | Distribution | Parameter | Distribution | Parameter | Distribution |
| **Poly degree** | Unif. Int. $[2, 10]$ | **C** | Unif. $[10^{-4}, 100]$ | **Num. Estimators** | Unif. Int $[10, 300]$ | **Max. depth** | Log Unif. Int $[2, 10]$ |
| **Alpha** | Log Unif. $[10^{-6}, 100]$ | **Epsilon** | Log Unif. $[10^{-5}, 100]$ | **Max. depth** | Log Unif. Int $[2, 20]$ | **Gamma** | Log Unif. $[10^{-5}, 100]$ |
| | | **Gamma** | Log Unif. $[10^{-5}, 1000]$ | | | **Eta** | Log Unif. $[10^{-5}, 0.99]$ |
| | | | | | | **Lambda** | Log Unif. $[10^{-5}, 1]$ |

Table 8: Number of training samples together with resulting $R^2$ scores of the trained ML models for the benchmark in Section 5.2.

| Benchmark | Dim. | $n_{\text{train}}$ | Test $R^2$ | | | |
|---|---|---|---|---|---|---|
| | | | RR | SVR | RF | XGB |
| Michalewicz | 2 | 200 | 0.69 | 0.89 | 0.8 | 0.98 |
| | 4 | 600 | 0.21 | 0.3 | 0.63 | 0.96 |
| | 8 | 1000 | 0.11 | 0.1 | 0.31 | 0.82 |
| Rosenbrock | 2 | 100 | 0.96 | 0.87 | 0.91 | 0.92 |
| | 4 | 200 | 0.96 | 0.83 | 0.74 | 0.84 |
| | 8 | 500 | 0.95 | 0.83 | 0.68 | 0.84 |

Otherwise, we used defaults specified in the `BoTorch` implementation of the `SingleTaskGP` class. We fit the GP model using `SciPy`'s implementation of the L-BFGS-B algorithm [Byrd et al., 1995] with 5 random restarts to maximize the log-marginal likelihood. If the model is not trained, new observations are incorporated by building a new GP model with updated data and the same hyperparameters as in the previous iteration (strategies for such are implemented in `GPyTorch`).

**Variational Gaussian Process Model.** For the SVGP model we use the implementation from `BoTorch` with 500 inducing points. We apply the same input and output transformations as with the GP implementation. As kernel, we found using the same five kernels as with GP provides to much overhead with SVGP. Therefore, we restricted the usage to the Matèrn 1/2 kernel. The model is fitted with 3 random restarts by maximizing the evidence lower bound via Adam [Kingma and Ba, 2015] (learning rate 0.1) combined with early stopping (patience 30) and the cosine annealing learning rate scheduler [Loshchilov and Hutter, 2017].

# G   EXPERIMENTAL DETAILS

Here, we present further details on the implementation of the conducted experiments.

## G.1   DETAILS BENCHMARK SECTION 5.2

For the benchmark, we trained various ML models ($f_{\text{M}}$) to be validated afterwards. We used RF, SVR, and RR as implemented in `Scikit-learn` [Pedregosa et al., 2011]. XGB is implemented based on [Chen and Guestrin, 2016]. Inputs and outputs were normalized to have zero sample mean and unit standard deviation for all models. Number of training samples and resulting test $R^2$ are given in Table 8.

**Hyperparameters.** For the Rosebrock benchmark, we used the specified default parameters to a large extend. Only exceptions are: **1)** RR with polynomial features of max. 3th degree with L2 regularization of 0.3 (alpha). **2)** RF with 200 trees (num. estimators) with maximum depth of 25. **3)** XGB with max. depth of 3. For the Michalewicz function, we found the need to tune the hyperparameters. Therefore, we used BO implemented in `Scikit-optimize` [Tim et al., 2021] based on the cross-validated mean absolute error objective. The hyperparameter searchspace is given in Table 7.

### G.2 BENCHMARK FUNCTIONS

This section gives an overview of the analytical functions used throughout this work.

**Series System with four branches.**   [Waarts, 2000]

$$f(x_1, x_2) = \min \begin{cases} 3 + 0.1(x_1 - x_2)^2 - \frac{x_1 + x_2}{\sqrt{2}} \\ 3 + 0.1(x_1 - x_2)^2 + \frac{x_1 + x_2}{\sqrt{2}} \\ (x_1 - x_2) + \frac{7}{\sqrt{2}} \\ (x_2 - x_1) + \frac{7}{\sqrt{2}} \end{cases}$$

where we used $x_i \in [-8, 8]$.

**Modified Rastrigin Function.**   [Törn and Zhilinskas, 1989]

$$f(x_1, x_2) = 10 + \sum_{i=1}^{2} \left( x_i^2 - 5 \cos 2\pi x_i \right)$$

where we used $x_i \in [-5, 5]$.

**Styblinski-Tang Function.**   [Styblinski and Tang, 1990]

$$f(\mathbf{x}) = 0.5 \sum_{i=1}^{d} \left( x_i^4 - 16 x_i^2 + 5 x_i \right)$$

where we used $x_i \in [-5, 5]$.

**Michalewicz Function.**   [Michalewicz, 1992]

$$f(\mathbf{x}) = - \sum_{i=1}^{d} \sin(x_i) \sin^{20} \left( \frac{i x_i^2}{\pi} \right)$$

where we used $x_i \in [0, \pi]$.

**Rosenbrock Function.**   [Rosenbrock, 1960]

$$f(\mathbf{x}) = \sum_{i=1}^{d-1} 100 \left( x_{i+1} - x_i \right)^2 + \left( x_i - 1 \right)^2$$

where we used $x_i \in [-2, 2]$.