# OpenReview forum: "Quantifying Local Model Validity using Active Learning"
_auai.org/UAI/2024/Conference — UAI 2024 poster_

### Official Review · Reviewer_Bzmn · 2024-03-07

**Q2-1 Originality-Novelty:** 2
**Q2-2 Correctness-Technical Quality:** 2
**Q2-5 Clarity Of Writing:** 2

**Q1 Summary And Contributions:**

This paper presents an active learning algorithm to select instances from a validation set for efficiently validating the error of a model on a subpopulation where instances suffer higher prediction biases. Experimental results demonstrate some results of the algorithm.

**Q2-3 Extent To Which Claims Are Supported By Evidence:**

2: Fair: the main claims are somewhat supported by evidence (but the experimental evaluation may be weak, or does not match entirely with the claims, important baselines may be missing, proofs contain important ideas but lack rigor, algorithmic details are only discussed superficially, references are imprecise, assumptions are not sufficiently motivated or explicated, etc.).

**Q2-4 Reproducibility:**

3: Good: key resources (e.g. proofs, code, data) are available and key details (e.g. proofs, experimental setup) are sufficiently well-described for competent researchers to confidently reproduce the main results.

**Q3 Main Strengths:**

S1. Model validation in the local region is an interesting and useful topic.

S2. The idea of introducing active learning to enhance local model validation is interesting.

**Q4 Main Weakness:**

W1. Presentation is hard to follow and confusing in many aspects.

W2. Research problem is not well articulated or justified.

W3. The work seems to largely overlap with well-known machine learning literature (especially active learning) but the connection and difference are not discussed.

**Q5 Detailed Comments To The Authors:**

[1] The research problem is not well articulated or justified. First, if our concern is model performance in a target sub-population $X_s$, why not simply train and evaluate the model based on data collected from $X_s$? Second, it makes sense to pre-define $X_s$ based on application e.g., we care about diagnosis accuracy on female patients. However, this work defines $X_s$ based on model error i.e., we want to evaluate the accuracy of a model on its mis-classified data -- this doesn't make much sense to me. (And my bold guess is, with mild assumptions, one can theoretically prove such an error depends on the given sub-population parameter $\xi$. Then, why bother to estimate the error at all?)

[2] The entire active learning (AL) literature is missing in discussion and comparison. This paper designs an error-based query algorithm, which is very common in AL (see B Settles's survey on AL for example). The connection must be discussed.

[3] The entire learning theory literature is missing in discussion and comparison. The present validation metric (1) is very common in machine learning theory; see e.g., the PAC framework.

[4] Design of the validation metric in Section 3 is confusing. Specifically,

-- In the definition of noiseless residual $\delta(x) = \mathbb{E}[\Delta(x)] = f_M(x) - f_E(x)$, it is not clear what is random in the expectation.

-- It is claimed  that $x$ is subject to uncertainty in RA, where this paper assumed no uncertainty. So does it mean $x$ is assumed fixed (and only noise is random? That sounds very odd e.g., how to interpret the tolerance inequality (1)? Is the probability defined based on the randomness of noise instead of $x$? If so, wouldn't both $\xi$ and $r$ be dependent on $x$? (But I think the analysis assumes it is universal.)

[5] Definition 2 of limit state seems quite restrictive. For a continuous domain, its probability measure is likely to be zero (e.g. the probability for a point in $\mathbb{R}^p$ to fall on a straight line is zero.) Some discussion would be helpful.

[6] In Section 4.2, it is stated this paper considers learning the limit state $S$. I don't understand what it means. Please clarify.

Overall, I think the paper has some good potential but the problem and methodology are not clearly presented and well justified. Also, my impression is that authors are not very familiar with machine learning literature. They are highly encouraged to have an ML expert help with reshaping and rewriting the work.

**Q9 Complying With Reviewing Instructions:**

Yes

---

> ### Author Rebuttal · Authors · 2024-04-06
>
> Thank you for your review! In the following, we address your main review and answer your questions. We hope these address your concerns and convince you to reconsider your score. If not, please let us know how we can clarify things further.
> ## Weakness & Comments
> **1. Research problem.** We are interested in validating the model within the **whole** domain $X$. The subdomain $X_s$ is unknown at the beginning, i.e. if we knew $X_s$ beforehand, the problem would have already been solved. Task is to *find* the sub-population $X_s$ where the model error is close to the tolerance $\xi$, so that we can derive from this the decision boundary between valid and invalid regions (see Figure 1). We will add a comment to make this more clear.
>
> **2. Active learning (AL) missing.** The term AL is used within different contexts, e.g. Settles survey focuses mainly on AL for classification, whereas in the RA domain, AL refers to algorithms for learning a limit state condition, as stated in Section 2. Latter is related to our proposed method. Similarities (Algorithm 1) can be found across AL (pool-based), BO and AL for RA. However, our strategy (MC-Prob) reduces the predicted misclassification probability derived from the GPs posterior in the *regression* setting (Section 4.4, p. 5). In contrast, error-based query algorithms described in Settles survey in Sec. 3.4 (p. 19), e.g. [1], use acquisition functions based on expected entropy or future error for the *classifier*. We will add a discussion of AL from ML domain in Section 2.
>
> Nevertheless, we performed an experiment for our analytical benchmark (Section 5.1) motivated by your comment. We tested the smallest margin method (Settles 2010, p. 13) with the XGB-Classifier (default hyperparameters), as this seems to be a good baseline [2, 3]. We report mean F1-Score across samples and 30 trials together with our method (Figure 3):
> ||Series|Rastrigin|Styb. 2d|Styb. 4d|Styb. 6d|Styb. 8d|
> |-|-|-|-|-|-|-|
> |Margin|0.67|0.64|0.61|0.23|0.12|0.06|
> |$\psi_{mis,0.2}$|**0.90**|**0.89**|**0.90**|**0.84**|**0.72**|**0.59**|
>
> We add the AL baseline to Figure 3 and description to Section 5.2.
>
> **3. Learning theory missing.** Our main contribution is based on the proposed AL strategy for validation (Section 4), and **not the development of a new validation metric**, as indicated by your comments. The validation metric (Eq. 1) was originally proposed by Rebba et al. [2008], as cited in the manuscript. Further, statistical learning theory focuses on solvability of problems with ML, while our specific focus lies in partitioning the input space into valid and invalid regions for a *trained* model.
>
> **4. Validation metric is confusing.** We will address questions separately:
>
> * **Randomness in the expectation.** $\Delta(x)$ is a Gaussian random variable (p. 3, above Eq. 1), defined as the difference between prediction and observation, and the label observation has additive Gaussian noise. Note, we use $\sim$ to denote “distributed as”. We will state this more clearly.
> * **Is $x$ assumed fixed.** We assume $x$ can be controlled and measured precisely (similar to Bayesian optimization), i.e. we have no uncertainty about the value we observe (see Section 3). We will state this more clearly.
> * **How to interpret Eq. 1.** It can be interpreted as the probability of the difference $\Delta(x)$ (at a point $x$) being within the selected tolerance level $\xi$, i.e. the probability of local validity of the model. The uncertainty is due to the label uncertainty (noise $\sigma^2_e$).
> * **Are $\xi$ and $r$ dependent on $x$.** $\xi$ and $r$ are independent of $x$, since they are chosen by the modeler/decision maker. To clarify, tolerance $\xi$ represents the margin of error the designer is willing to accept in the predictions. Probability $r$ is the confidence level or the likelihood that any new observation falls within this predefined tolerance interval.
>
> **5. Definition 2 seems restrictive.** We defined the limit state $S$ to be the set of points, where the **noiseless** absolute difference between prediction and observation is equal to the tolerance level (Definition 2, p. 4). Additionally, $S$ can be seen as the set of points on the boundary between the valid and invalid domains (see Figure 1). We will provide further clarification on Definition 2 in the final text.
>
> **6. Limit state.** The limit state $S$ is unknown in practice. Moreover, our objective is to construct a strategy aimed at placing samples in the vicinity of $S$, thereby efficiently learning to differentiate between valid and invalid regions of the model under validation.
> ## References
> [1] N. Roy, et al. Toward optimal active learning through sampling estimation of error reduction. ICML 2001.
>
> [2] G. Cawley. Baseline methods for active learning. Workshop, AISTATS 2010
>
> [3] Y. Yang, M. Loog. A benchmark and comparison of active learning for logistic regression. Pattern Recognition, 2018.

---

### Official Review · Reviewer_M6CK · 2024-03-21

**Q2-1 Originality-Novelty:** 2
**Q2-2 Correctness-Technical Quality:** 3
**Q2-5 Clarity Of Writing:** 2

**Q1 Summary And Contributions:**

This manuscript introduces an approach for quantifying the local model validity in machine learning using active learning where accurate validation is crucial. The proposed approach focuses on determining where the model's error stays within a predefined tolerance level, so that can control the data needed for validation, it is implemented by defining the local validity and employing Gaussian Process regression. The proposed active learning acquisition function is designed based on misclassification probability during the sampling process.

**Q2-3 Extent To Which Claims Are Supported By Evidence:**

2: Fair: the main claims are somewhat supported by evidence (but the experimental evaluation may be weak, or does not match entirely with the claims, important baselines may be missing, proofs contain important ideas but lack rigor, algorithmic details are only discussed superficially, references are imprecise, assumptions are not sufficiently motivated or explicated, etc.).

**Q2-4 Reproducibility:**

2: Fair: key resources (e.g. proofs, code, data) are unavailable but key details (e.g. proof sketches, experimental setup) are sufficiently well-described for an expert to confidently reproduce the main results.

**Q3 Main Strengths:**

This work is complete, the authors provide a solid theoretical framework for the proposed active learning based on local model validity with the limit state, and the development of a novel acquisition function based on misclassification probability.

**Q4 Main Weakness:**

1. The tolerance level heavily affects the performance of the model.

2. The assumption of homoscedastic noise might not fit the real-life scenarios, noise can also be heteroskedastic.

3. The computation cost of GP-based model is heavily affected by the dataset size.

**Q5 Detailed Comments To The Authors:**

1. Can the model be applied to very high-dim data?

**Q9 Complying With Reviewing Instructions:**

Yes

---

> ### Author Rebuttal · Authors · 2024-04-04
>
> Thank you for your constructive and detailed feedback! In the following, we address your review and answer your questions. We hope these address your concerns and convince you to reconsider your score. If not, please let us know so we can clarify things further.
> ## Weakness
> **1. Tolerance level affects performance.** This is a good point! The tolerance level can impact the performance of the model. With a stricter tolerance (resulting in more invalid regions), the problem can become more challenging to model, except in cases where everything is invalid, making the problem much easier. However, note this is a **property of the problem** and not a property of our method. In our benchmark study (Section 5.2, p. 7) we have a diverse set of examples, by maintaining a fixed tolerance while using different models, which may perform well or poorly. The reason behind using Bayesian optimization in Section 5.2 for the Michalewicz benchmark is to obtain models that are at least partially valid (otherwise the problem would be easy as stated above). For example, SVM would be completely invalid for most examples without some hyperparameter tuning.
>
> However, to take different tolerances into account, we performed a further ablation study by evaluating the adaptive methods with different tolerance levels leading to different ratios (valid/all) for the 4d Styblinsky-Tang function. Other settings were kept the same as in Section 5.1. We report the results for the mean F1-Score across samples and 30 trails in the following:
> ||20% valid|40% valid|60% valid|80% valid|
> |-|-|-|-|-|
> |$\psi_{mis,0.2}$|**0.80**|**0.88**|**0.92**|**0.96**|
> |$\psi_{U}$|0.76|0.87|0.91|0.95|
> |random|0.75|0.86|0.91|0.95|
>
> The F1-score drops with stricter tolerance (e.g. 20% valid). Nevertheless, all methods can handle this reasonably well. We will include the ablation study in the final version as plots (which we cannot include here due to restrictions) and comment on the impact of the tolerance.
>
> **2. Heteroskedastic noise and real datasets.** Yes, real-world problems can be heteroskedastic, and GP models can be extended for this setting (see Appendix B, p. 13). Nevertheless, homoscedasticity can be a reasonable assumption, even for real world data, as it is frequently assumed in the Bayesian optimization literature (e.g. [1, 2, 3]).
>
> To extend our benchmarks and demonstrate that our method is applicable to real-world data, we conducted further evaluations using a tabular dataset from OpenML (TaskID 4835, 6D). We used similar settings as in Appendix E.1, with a GP instead of SVGP, and 621 train, 1243 validation and 1243 test samples. Two models were validated, a GP (Matern 5/2) and SVM (default hyperparameters), with a tolerance of $\xi=0.1$. We report the mean and final results across samples for 20 trails, and across GP ($\sim$64% valid) and SVM ($\sim$65% valid) with 60 initial and 800 adaptive samples:
> |||mean|||final|||
> |:-:|:-:|:-:|:-:|:-:|:-:|:-:|:-:|
> ||$\psi_{mis,0.2}$|$\psi_{U}$|random|$\psi_{mis,0.2}$|$\psi_U$|random|**all samples**|
> |F1-Score|**0.786**|0.783|0.781|**0.795**|0.793|0.789|0.798|
> |Precision|**0.680**|0.674|0.670|**0.698**|0.695|0.689|0.730|
> |Recall|0.933|**0.937**|**0.937**|0.922|**0.925**|**0.925**|0.881|
>
> We will include the results as plots in Appendix E.1.
>
> **3. Computational cost of GP.** Especially for low sample settings, such as our target application, GP models have proven to be very useful, with computational cost playing a minor role. Nevertheless, recent advances in GP regression allow exact GPs to scale up to larger datasets without sacrificing accuracy (see, for instance, [4]). If computational expense is an issue, one can get around by using approximate GP models [Hensman et al.,2013], which have a fixed number of inducing points. However, this is beyond the scope of our current work. In our introduction, we will clarify further that our target application is a low sample setting.
> ## Comments
> **Very high-dim data.** We will add further discussion around the limitation of our method (Appendix B, p12) and make it clearer that our model has difficulty handling very high dimensional problems due to using a GP with distance based kernel. This is a trade-off required for being able to handle small data settings well.
> ## References
> [1] M. Neumann-Brosig, A. Marco, D. Schwarzmann and S. Trimpe, "Data-Efficient Autotuning With Bayesian Optimization: An Industrial Control Study," in IEEE Transactions on Control Systems Technology, vol. 28, no. 3, 2020.
>
> [2] Berkenkamp, F., Krause, A. and Schoellig, A.P. Bayesian optimization with safety constraints: safe and automatic parameter tuning in robotics. Mach Learn 112, 2023.
>
> [3] B. Letham. B. Karrer. G. Ottoni. E. Bakshy. "Constrained Bayesian Optimization with Noisy Experiments." Bayesian Anal. 14 (2), 2019.
>
> [4] J. Gardner, G. Pleiss, K. Q. Weinberger, D. Bindel, and A. G. Wilson, “GPyTorch: Blackbox Matrix-Matrix Gaussian Process Inference with GPU Acceleration,” NIPS, 2018.

---

### Official Review · Reviewer_btVa · 2024-03-22

**Q2-1 Originality-Novelty:** 3
**Q2-2 Correctness-Technical Quality:** 3
**Q2-5 Clarity Of Writing:** 3

**Q1 Summary And Contributions:**

This paper is about quantifying the validity of a model, locally, using active learning.
AL is used to construct the points to compute the model error, then to construct the test set to focus on local error.
Model validation is done uner local validity.
A gaussian process is used to measure the belief in the limit state.

**Q2-3 Extent To Which Claims Are Supported By Evidence:**

3: Good: the main claims are supported by convincing evidence (in the form of adequate experimental evaluation, proofs, (pseudo-)code, references, assumptions).

**Q2-4 Reproducibility:**

2: Fair: key resources (e.g. proofs, code, data) are unavailable but key details (e.g. proof sketches, experimental setup) are sufficiently well-described for an expert to confidently reproduce the main results.

**Q3 Main Strengths:**

- formalism is ok
- results are interesting
- experiments illustrate the method

**Q4 Main Weakness:**

- the bound may be more discussed
- the experimental section may be clearer

**Q5 Detailed Comments To The Authors:**

this paper is very interesting, but this is not my domain, so i preferred to focus on the other papers i have to review.
sorry for this!

**Q9 Complying With Reviewing Instructions:**

Yes

---

> ### Author Rebuttal · Authors · 2024-04-04
>
> We thank you a lot for your positive feedback! Below are our answers to your questions and comments. We are eager to discuss things further if necessary.
> ## Weakness
> **1. Error bounds more discussed.** Appendix C.1 briefly discusses the error bound using the 90% confidence interval of the GP and the error bounds based on Theorem 1 (see Figure 5 and Figure 6). We will extend this section. In particular, we will further explain that the true error is estimated by the interval with a 90% confidence level individually for each point of the input space. This bound directly uses the learned uncertainty of the GP model, which may underestimate the error, but in practice provides a good bound in most cases. In contrast, Theorem 1 states that the true error is less than the error bound $\eta$ **for all points in the input space** with a probability of 90%. The result gives a uniform error bound that is not directly dependent on the accuracy of the predicted uncertainty of the GP, and is therefore much stronger. However, the theorem requires additional knowledge on the Lipschitz continuity of the covariance kernel as well as $\delta$ and is therefore not generally applicable. Furthermore, the figures show that the bound may be too conservative, especially when using estimated Lipschitz constants. Additionally, we would like to point out that our sampling method results in a rapid decrease of the error bound $\eta$ near the limit state even for small sample sizes (see Figure 5).
>
> **2. Experimental section.** Thanks for pointing this out! We could make this section more accessible by 1) provide reasoning why we do these experiments and 2) are the methodology and metrics justified. Would that be helpful?

---

### Official Review · Reviewer_YDvH · 2024-03-22

**Q2-1 Originality-Novelty:** 2
**Q2-2 Correctness-Technical Quality:** 2
**Q2-5 Clarity Of Writing:** 3

**Q1 Summary And Contributions:**

The paper proposes a method for assessing the validity of machine learning models in real-world applications, considering legal and policy regulations. These regulations often require the model's approximation error to be below a certain threshold. While a global metric is not sensitive enough for assessing individual predictions, local validity evaluation is costly due to the need for additional data. The proposed solution involves learning the model error to estimate validity locally while reducing data requirements through active learning. Empirical evidence from model validation benchmarks supports the effectiveness of this method, showing that it can generate an error model with sufficient discriminative properties using a relatively small amount of data. Additionally, the proposed approach demonstrates increased sensitivity to local changes in validity bounds compared to alternative methods.

**Q2-3 Extent To Which Claims Are Supported By Evidence:**

2: Fair: the main claims are somewhat supported by evidence (but the experimental evaluation may be weak, or does not match entirely with the claims, important baselines may be missing, proofs contain important ideas but lack rigor, algorithmic details are only discussed superficially, references are imprecise, assumptions are not sufficiently motivated or explicated, etc.).

**Q2-4 Reproducibility:**

2: Fair: key resources (e.g. proofs, code, data) are unavailable but key details (e.g. proof sketches, experimental setup) are sufficiently well-described for an expert to confidently reproduce the main results.

**Q3 Main Strengths:**

The paper is nicely written, extensive containing both theory and empirics.

**Q4 Main Weakness:**

It is unclear to me how relevant the methodology put forward here is but admittedly this paper is outside my comfort zone.

**Q5 Detailed Comments To The Authors:**

It would have been great to have a bit more context in the illustrations

**Q9 Complying With Reviewing Instructions:**

Yes

---

> ### Author Rebuttal · Authors · 2024-04-04
>
> We thank you a lot for your positive feedback! Below are our answers to your questions and comments. We are eager to discuss things further if necessary.
>
> ## Weakness
> **Relevance of methodology.** Methods like ours are vital for increasing the integration of machine learning models into safety-critical devices, such as virtual sensors integrated into passenger cars [1]. Since these systems must comply with national or international regulations (e.g., ISO 262626), rigorous validation is necessary but often of high cost and effort. Our work is relevant and motivated, as there is currently very little comparable research. While methods like conformal prediction exist, our approach offers distinct advantages, particularly in scenarios with limited data available (see Appendix D, p. 14). Therefore, we believe our method fills a gap in research by providing a strategy to validate such models efficiently.
>
> ## Comments
> **More context of illustrations.** Thank you for pointing out further improvements! We are happy to add more context to our illustrations.  For example, in Figures 5 and 6, we will incorporate the additional details as described in the response to Reviewer btVa. Could you please elaborate on the context and figures we should further address?
>
> ## References
> [1] W. Kirchgässner, O. Wallscheid and J. Böcker, "Estimating Electric Motor Temperatures With Deep Residual Machine Learning," in IEEE Transactions on Power Electronics, vol. 36, no. 7, pp. 7480-7488, July 2021, doi: 10.1109/TPEL.2020.3045596.

---

### Official Review · Reviewer_q8ub · 2024-03-30

**Q2-1 Originality-Novelty:** 3
**Q2-2 Correctness-Technical Quality:** 3
**Q2-5 Clarity Of Writing:** 2

**Q10 Ethical Concerns:**

NA.

**Q1 Summary And Contributions:**

This paper defines the notion of "local model validity" as a model's prediction not deviating from the ground truth  (in either direction) by more than a predefined threshold (also referred to as the *limit state*). It then proposes to identify regions of local validity for a model by learning a companion Gaussian Process model via Bayesian Optimization (BayesOpt). Although it extends ideas from the reliability literature where similar problems have been studied, the paper makes it own novel contributions, e.g., the general framing and acquisition function for BayesOpt.

**Q2-3 Extent To Which Claims Are Supported By Evidence:**

4: Excellent: all claims are supported by very convincing evidence (in the form of comprehensive experimental evaluation, rigorous mathematical proofs, detailed (pseudo-)code, precise references, well-motivated and realistic assumptions) and the authors deliver what they promise.

**Q2-4 Reproducibility:**

2: Fair: key resources (e.g. proofs, code, data) are unavailable but key details (e.g. proof sketches, experimental setup) are sufficiently well-described for an expert to confidently reproduce the main results.

**Q3 Main Strengths:**

Rigorous in its approach to a solution.

**Q4 Main Weakness:**

Writing can be clearer wrt both flow and notation.

**Q5 Detailed Comments To The Authors:**

Comments:
* I think the paper solves an interesting problem, and although similar problems have been solved in the reliability literature, the extension here is non-trivial.
* In general notation can be cleaner and flow might be improved:
  *  Its confusing to refer to the difference in the response values as $\Delta x$.
  *  Do we need two different sets of notations for the GP: $\mu_{y|D}$ and $\mu_{g|D}$, and similarly for $\sigma$? It is usually understood from context when the distribution over functions vs that over response is intended.
  *  If the function we're interested in $\hat{g}$, shouldn't the subscript reflect that in $\mu_{g|D}$, i.e., should be $\mu_{\hat{g}|D}$?
  *  $\theta$ is suddenly introduced in the "Learning Hyperparameters" subsection. It would help with readability to provide examples of $\theta$, e.g., maybe this is something like the *length scale* in the RBF kernel?
  * In the "Validation Metric" subsection in Section 3, $r$ makes a brief appearance and then isn't used subsequently.
  * in Definition 1, in this snippet "....if $\xi - |\delta(x)| > 0$", should the relation be $\geq$?
  * In the "Observations" subsection in Section 3, $p(\epsilon')$ should probably be $p(\epsilon)$.
  * The first para on Page 3 talks about some of the contributions - it might be better to group them in the "Contributions" subsection in Section 1.
  * In Section 4.3, please introduce the "composition" notation $\circ$.
  * Its good for readability to impart some intuition alongside introducing a formula. Take the example of Eqn. (4) in Section 4.4 - why does this acq. fn. make sense?
  * In general, I thought that the problem's framing could be made simpler. The key task seems to be two learn two level sets, one each for $-\xi$ and $\xi$; it might be simpler to motivate the problem as such.
* In the "Experiments" section, what does it mean when you say "Even if the model is not retrained, we update the GP...."? Does training here refer to learning $\theta$? If so, please make this clear.
* Here's probably a silly question: one point of difference from RA highlighted is that we have two limit states here. What if we run a RA technique twice, once for $-\xi$ and $\xi$ each? How does the proposed technique then compare? If the authors also think this is a simple practical "solution" they should account for this in their motivation.
* The authors also mention that one difference relative to RA is we assume response noise - but irrespective of whether this is assumed in RA or not, surely techniques such in Bichon et. [2008], are *resilient* to some noise just because they're using GPs? If so this nuance should be mentioned.
* If the problem boils down to learning level sets, would something like *InfoBAX* [1] work too? To be clear, I don't want authors to compare against InfoBAX (since this is fairly recent), but I would request them to cite it in future versions of the paper, if they find it relevant.
* In the "Introduction", the authors mention "Since they often do not include further domain-specific inductive biases such as the underlying physics...". This is not true for *Physics Informed Neural Networks (PINNs)* and should probably be cited.
* Reg. the experiments for Table 3: I'm curious as to why trying out so many classifiers, e.g., RR, SVR, etc, is important. We are just modeling the difference in prediction vs ground truth.

Minor:
* Page 8, typo "trails".


[1] https://willieneis.github.io/bax-website/

**Q9 Complying With Reviewing Instructions:**

Yes

---

> ### Author Rebuttal · Authors · 2024-04-05
>
> Thank you very much for your constructive and detailed feedback! We hope to address your concerns and convince you to raise your score. If not, please reply to this post, and we'll gladly provide further clarification.
>
> ## Weakness & Comments
> **1. Notation can be cleaner.** Thank you for pointing out! We fixed most of the points and will address remaining ones if not stated otherwise. We comment on some suggestions in the following:
>
> * **Subscript $\hat{g}$.** $\hat{g}$ is the surrogate that models the limit state function $g$, i.e. function of interest. Furthermore, $\hat{g}$ outputs a distribution $p(g|x,D)$, with mean $\mu_{g|D}$ and variance $\sigma^2_{g|D}$. We use $\sim$ throughout the paper to denote “distributed as”. We will state this clearly in the text.
>
> * **Different set of notations.** The difference can be nuanced but is necessary to avoid ambiguity. $\mu$ and $\sigma^2$ with subscript $y$ describe GP mean and var. *before* transformation with $\lambda$. Subscript $g$ is used *after* transformation, i.e. parameters of a folded Gaussian (see Sec. 4.3).
>
> * **Appearance of $r$ in Sec. 3.** Yes, $r$ appears once to be consistent with its original paper. In Sec. 4, Eq. 7 includes both epistemic unc. from the GP model and aleatoric unc. from the estimated noise. $r$ is only related to aleatoric unc. from the label noise. We will remove $r$ and provide a description instead.
>
> * **Intuition alongside formula.** By using the acquisition function (AQ) in Eq.4, we select samples which have large variance and are close to the limit state according to the GP. $|\mu_{g|D}|$ is small only if $x$ is close to the limit state. Placing samples in close proximity of the limit state improves the GP's ability to distinguish between valid and invalid regions, similar to RA. We incorporate your suggestion and provide further insights.
>
> * **Framing as two level sets.** Good point! RA is very similar to the task of level set learning. We will include motivation also from this perspective.
>
> **2. Updating GP.** To reduce computational effort, we retrain hyperparameters $\theta$ such as length scale and kernel variance only in some iterations and keep them fixed in subsequent ones. Hence, we build a new GP model with updated data and fixed hyperparam. Further performance improvements could be made by using faster updating strategies instead of building the new GP, e.g. [1], which we have not used. We will clarify and add further comments.
>
> **3. Run RA twice.** Interesting point! This would lead to two GPs ($\xi-\Delta$, $\xi+\Delta$) and two AQs. We tested this approach for the analytical benchmark (Sec. 5.1) with U-Fun (Echard,2011) and sum both functions ($\psi_{U,double}$). The performance is similar to MC-Prob as (almost) expected, since the U-Fun minimizes the misclassification probability in RA. However, we have two GPs, which add to the computational cost and are more prone to misspecification. We report the mean F1-Score across samples and 30 trials:
> ||Series|Rastrigin|Styb. 2d|Styb. 4d|Styb. 6d|
> |-|-|-|-|-|-|
> |$\psi_{mc}$|0.894|**0.889**|0.889|0.831|0.724|
> |$\psi_{mc,0.2}$|**0.900**|0.888|**0.901**|**0.839**|0.719|
> |$\psi_{U,double}$|0.892|0.831|0.898|0.835|**0.733**|
>
> We will include this result together with missing Styb. 8d result (currently running) in Figure 3 and add discussion around this.
>
> **4. RA with noise.** Good point! If we would use a GP model with RA techniques, we could also handle noisy observations to some extent. However, in RA this topic became only very recently relevant, e.g. see [2, 3]. We will add a comment to account for this.
>
> **5. Level Sets.** Interesting paper! InfoBAX proposes to maximize entropy gained from including a sample in the model wrt. some arbitrary algo. output, which our problem could potentially framed as such. However, the approach seems computationally very expensive; to evaluate the AQ we have to create and evaluate multiple GP models for the $l$ execution path samples (see Eq. 4 & 16 in the InfoBAX paper). Nevertheless, this seems interesting and we will cite it.
>
> **6. Inductive Bias.** We agree, PINN or other approaches can incorporate prior knowledge. We will clarify that, e.g. PINNs are capable of including such knowledge, and cite relevant resources, e.g. [4].
>
> **7. Many models in experiments.** Error surfaces of models (e.g. SVR, RF) can be different, e.g. SVR is smoother than RF. We found latter to be more difficult to validate since tree-models could be invalid in small regions (w.r.t. input domain) due to the learned decision surface (mentioned briefly in Appendix B). We will add further information in Sec. 5.2.
>
> ## References
> [1] S. Stanton, et al., Kernel Interp. for Scalable Online GPs. AISTATS 2021.
>
> [2] J. Chun. Active Learning-Based Kriging Model with Noise Responses. Appl. Sci. 2024.
>
> [3] A. Pires, et al. Rel. Ana. for data-driven noisy models using active learning, arXiv:2401.10796.
>
> [4] M. Raissi, et al. Physics-informed NN, Jour. Comp. Phys., 2019.

---

### Meta-Review · Area_Chair_aVfs · 2024-04-15

The five reviewers are in favour of accepting this paper. The original reviews pointed out strengths and some weaknesses of the work, and suggested improvements. After the rebuttal by authors the reviewers remained generally in favour of this paper, but there were some remaining concerns. If this paper was to be accepted, I would request the authors to respect the promises made in the rebuttal, including the clarifications and discussions, additional references, as well as fixing the issues that were pointed out in the reviews. I would encourage the authors to do this conscientiously in that case.

I would like to add these requests: Check the formatting of the references, to ensure complete and consistent venue names. Some references looked incomplete (e.g. Loshchilov and Hutter 2017, and the first two references in the list). Last but not least, I encourage the authors to carry out a through proof reading.